# Bioactivity of the Genus *Turnera*: A Review of the Last 10 Years

**DOI:** 10.3390/ph16111573

**Published:** 2023-11-07

**Authors:** Aída Parra-Naranjo, Cecilia Delgado-Montemayor, Ricardo Salazar-Aranda, Noemí Waksman-Minsky

**Affiliations:** Facultad de Medicina, Departamento de Química Analítica, Universidad Autónoma de Nuevo León, Monterrey 64460, Nuevo León, Mexico; aida.parranr@uanl.edu.mx (A.P.-N.); cdelgado.me0018@uanl.edu.mx (C.D.-M.); ricardo.salazarar@uanl.edu.mx (R.S.-A.)

**Keywords:** *Turnera*, damiana, bioactivity, toxicity, aphrodisiac, hepatoprotection, antioxidant, ethnomedicine

## Abstract

*Turnera* is a genus of plants whose biological activity has been widely studied. The importance of this genus, particularly *Turnera diffusa*, as a source of treatment for various conditions is evidenced by the large number of new studies that have evaluated its biological activity. Accordingly, the objective of this review was to compile the information published in the last ten years concerning the biological activities reported for *Turnera* spp. The present work includes 92 publications that evaluate 29 bioactivities and toxicological and genotoxic information on five species of this genus. Among the pharmacological effects reported, the antioxidant, hepatoprotective, neuroprotective, hypoglycemic, and aphrodisiac activities seem more promising. Phytochemicals and standardized plant extracts could offer alternative therapeutic remedies for various diseases. Although several flavonoids, cyanogenic glycosides, monoterpenoids, triterpenoids, and fatty acids have been isolated for *Turnera* plants, future research should focus on the identification of the main active principles responsible for these pharmacological activities, as well as to perform clinical trials to support the laboratory results.

## 1. Introduction

*Turnera* L. is a genus of plants, of which around one hundred and twenty-eight species of American origin and two of African origin have been reported [1]. In the Americas, it is distributed in semi-arid areas of Mexico, Central America, the Caribbean islands, and South America [2].

*Turnera diffusa* Willd. ex Schult. (Turneraceae), generally known as damiana, is the most representative species of its genus. In a PubMed search of publications covering the last 10 years, 92 articles referring to *Turnera* were found. The same number was found for *Turnera diffusa*, while in second and third place were *Turnera subulata* Sm. (Turneraceae) (16 mentions) and *Turnera. ulmifolia* L. (Turneraceae) (10 mentions), respectively.

*T. diffusa* has been part of folk medicine since ancient times and was considered to be one of the most important therapeutic agents in ancient Mayan civilization [3].

In a previous review, ethnobotanical and phytochemical information on the plants of this genus was collected and categorized as well as details of their bioactivity [4]. Since then, much new scientific information has appeared, so the objective of the present revision is to collate the experimental data published in the last 10 years (2013–2023) that reveal some of the biological activities reported for the plants of the genus, with particular emphasis on *T. diffusa*. Several biological actions such as adaptogenic as well as gastroprotective activities have not been included in this review, considering that we did not find new information published after the abovementioned revision.

The name for the species mentioned in each report corresponds to that assigned by the authors of each original article. However, when the name does not coincide with the species included in the classification proposed by Rocha et al. [1], the species to which the synonym corresponds is indicated in parentheses. Synonyms used in the text were corroborated with the Plants of the World database [5].

We also conducted a search for clinical trials performed with *Turnera* spp. in the Cochrane Library database. Six clinical trials were found, although four of them were already addressed in the previous review on *Turnera* spp. [4] Since the two remaining clinical trials aimed to evaluate the effects of the consumption of multivitamins (which included *T. diffusa* among its ingredients) with over 35 components on mood and cognitive abilities, we decided to leave them out for two reasons. Firstly, the evaluated effect did not correspond to any of the bioactivities addressed in the present review. Secondly, because the multivitamins evaluated contained a large number of components, it would not be very easy to associate any observed effects with the presence of *T. diffusa*.

## 2. Search Strategy

A bibliographic search was performed using the terms “*Turnera*” OR “*Turnera diffusa*” in the following databases: PubMed, Reaxys, Scopus, Web of Science, Sciencegate, and Cochrane Library. Language filters were used, selecting only those entries in English, Spanish, Portuguese, or German. In addition, in all databases except PubMed, the article and meeting abstract/conference paper filters were applied. The bibliographic search covered articles published up to 2023. We also included in the present revision those publications prior to 2014–2023 that have not been reported in the previous review of *Turnera* spp. [4] as well as the results of any critical bioactivity that was omitted in it.

## 3. Results

Considering that the activities reported for the plants of the genus, which we included in this review, are many and very diverse, we have considered first describing those with the most significant number of reports, such as antioxidant, anti-inflammatory, and immunomodulatory activities. Then, we mention the activities related to organ protection and the activities related to metabolic syndrome. Later, we reviewed their stimulant or energizing properties, including aphrodisiac activity (one of its most popular uses), followed by reports of enzymatic inhibition and their activity on blood vessels. Towards the final part, we addressed the antiparasitic, antimicrobial, and toxicity-related activities.

### 3.1. Antioxidant Activity

Under normal conditions, the production and elimination of reactive species of oxygen (ROS) and nitrogen are in balance, thanks to the antioxidant defense system. However, when alterations cause the overproduction of these species, an imbalance occurs in the organism, and a state of oxidative stress is established. Oxidative stress has been shown to strongly affect many diseases and conditions, such as cancer, inflammation, and neurodegenerative diseases. One of the therapeutic strategies currently being explored is using antioxidant substances that help combat oxidative stress [6].

Wong et al. compared the performance of phenolic compounds obtained from *T. diffusa* stems and leaves by two extraction techniques and with different concentrations of solvents. Although no statistically significant differences were obtained, the reflux extraction technique with heat and a 35% ethanol concentration step allowed for obtaining a higher yield of total phenolic compounds than other methods. In addition, the ability to eliminate free radicals was evaluated using the 2,2-Diphenyl-1-picrylhydrazyl (DPPH) method; an inhibition percentage of 77.9–79.7% was obtained when the extract was evaluated at 1 mg/mL [7].

Similarly, Wong et al. found no significant differences in the yield of total phenolic compounds when extracting *T. diffusa* leaves and stems by a microwave-assisted extraction compared with a heat reflux extraction [8].

Soriano et al. studied the effect of ultraviolet (UV) radiation exposure on the antioxidant content in *T. diffusa* leaves in an in vitro model. No significant differences in superoxide dismutase (SOD) enzyme activity were found between UV-C-irradiated plants compared with the non-irradiated group. However, there was a significant decrease in total peroxidase (POX) activity [9]. On the other hand, when plants were exposed to UV-B radiation, there was a significant difference in SOD activity but not in POX activity. The authors report that although low exposure for a short period stimulates the production of phenolic content in the plant, excessive irradiation decreases the phenolic content and the antioxidant activity evaluated by the DPPH method [10].

Wong et al. reported that the hydroalcoholic extract of the leaf and stem of *T. diffusa* obtained by a heat reflux extraction presents a good capacity as a free radical scavenger with the DPPH method (76.03% inhibition). At the same time, the activity decreases to 30% when the 2,2′-azino-bis(3-ethylbenzothiazoline-6-sulfonic acid) (ABTS) method is used and to 65% with the lipid oxidation inhibition assay [11]. When an ultrasound-assisted extraction was applied, a DPPH scavenging of 62% was obtained, while for the lipid oxidation inhibition assay, the result was of 23% [12].

Stashenko et al. reported that in the Oxygen Radical Absorbance Capacity (ORAC) assay, the essential oil (EO) of leaves and stems of *T. diffusa* have an antioxidant capacity of 813 µmol Trolox/g sample [13].

Kalimuthu et al. tested the antioxidant activity of ethanolic and methanolic extracts of *T. ulmifolia* callus by DPPH and ferric reducing antioxidant power assays (FRAP). Their results showed similar activity for both extracts, exhibiting dose-dependent antioxidant activity [14].

Urbizu et al. compared the antioxidant content of aqueous extracts from the aerial part of *T. diffusa* collected from different sites in Tamaulipas, Mexico. They found a correlation between the content of phenolic compounds and the antioxidant activity determined by the ABTS method [15].

According to Esquivel et al., the daily administration of 150 mg/kg of the hydroalcoholic extract of the aerial part of *T. diffusa* for 5 weeks reduces the levels of malondialdehyde (MDA) and nitric oxide (NO) in the mitochondria of rats with streptozotocin-induced diabetes [16].

Pio et al. found differences in the total phenolic content and antioxidant activity between *T. diffusa* var. *diffusa* and *T. diffusa* var. *aphrodisiaca* using the ABTS method [17].

Tsaltaki et al. compared the extraction efficiency of phenolic compounds from different species using conventional and microwave- and ultrasound-assisted extraction methods. They found that, among the plants evaluated, *T. diffusa* had the highest level of total phenolic compounds and the best antioxidant activity, as assessed by the DPPH method. Although all the extraction methods evaluated gave good yields, the highest amount of total phenolic compounds was obtained with the microwave-assisted extraction, giving a concentration of 395 mg of gallic acid per gram of dry weight. In contrast, the conventional extraction generated the highest antioxidant activity, with 377.21 mg of trolox per dry weight [18].

Aguirre et al. reported that, although the methanolic leaf extract of *T. diffusa* has great potency as an antioxidant, with an EC_50_ of 28.9 µg/mL in the DPPH assay, it has low efficacy as a free radical scavenger, with an E_max_ of 34.9%. On the other hand, in the bleaching test of β-carotene, *T. diffusa* did not show significant activity [19].

The antioxidant activity of *Turnera* spp. has been widely reported and has been found to be related mainly to the flavonoid content present in this genus [4]. Table 1, Table 2, Table 3, Table 4 and Table 5 show the reports in the literature that have evaluated the antioxidant activity of *Turnera* spp., either by biochemical or radical methods, in vivo models, or others.

The relationship between oxidative stress and chronic degenerative diseases such as diabetes, neurodegenerative diseases, and chronic inflammation, among others, is well described [6]. The study of the antioxidant activity of *Turnera* spp. extracts may contribute to elucidating part of the mechanism of action through which these exert biological activities, such as anti-inflammatory, hepatoprotective, or neuroprotective activities, to mention a few. Table 5 includes mostly studies whose main objective was not to evaluate antioxidant activity. However, these studies show the ability of the extracts to modulate the activity of antioxidant enzymes [28], in addition to decreasing the levels of lipid peroxidation and cell damage caused by the generation of free radicals and reactive oxygen species [27,30]. Therefore, Table 1, Table 2, Table 3, Table 4 and Table 5 may include results not discussed in this section but will be addressed later. In summary, the antioxidant activity of *Turnera* spp., which is one of the most reported in its traditional use, has been widely proven; the results of publications in recent years support the antioxidant characterization of this genus.

### 3.2. Anti-Inflammatory Activity

Inflammation occurs as a response of the organism to the presence of pathogens or injuries. It is a complex process in which different immune system cells participate and involves the secretion of different mediator molecules, such as prostaglandins, cytokines, and ROS, among others [31].

In their review, Szewczyk et al. [4] reported on the anti-inflammatory activity of *T. ulmifolia* with the carrageenan-induced edema model in rats. Similar results were obtained by Lawal et al., who evaluated the anti-inflammatory activity of EO from the leaves and stem bark of *T. diffusa* using the same model. The EO (100–400 mg/kg) from the stem bark significantly reduced the edema generated for up to 4 h after administration. At the same time, the EO from the leaf only did so up to 3 h post-administration [32].

Furthermore, according to Rebouças et al., the hydroalcoholic extract of *T. subulata* flowers has anti-inflammatory activity since, at a dose of 40 mg/kg, it is capable of significantly reducing abdominal edema induced by carrageenan in a model with *Danio rerio* [24].

Additional reports have emerged evaluating the anti-inflammatory activity of *Turnera* spp. using in vitro models. Cabral-Souza et al. evaluated the anti-inflammatory activity of an aqueous extract of *T. subulata* leaves on the RAW264.7 macrophage cell line stimulated with lipopolysaccharide (LPS). The co-treatment of LPS and the extract (500 µg/mL) inhibited ERK-1/2 phosphorylation, but not p38 and SAPK/JNK. In addition, CD40 and the receptor for advanced glycation end products, but not TLR4, whose expression is modulated by LPS in the inflammatory process, were found to be significantly decreased in co-treatment with *T. subulata*. This effect was also observed in the secretion of TNF-α and IL-1β, which participate in the inflammation process [33].

Saravanan et al. evaluated the anti-inflammatory potential of *T. subulata* using two models. The chloroform and ethanol extracts had good protein denaturation inhibition potential, with an IC_50_ of about 80 µg/mL, while that of aspirin, used as a positive control, was 63 µg/mL. On the other hand, in the erythrocyte membrane stabilization assay, the chloroform extract had an IC_50_ of 74 µg/mL and the ethanol extract of 77 µg/mL, values similar to the IC_50_ of aspirin (68 µg/mL) [21].

The results of Lawal et al. [32] suggest that the components that may be present in stem bark EO, in contrast to the components present in *T. diffusa* leaf EO, contribute to a more sustained anti-inflammatory effect. On the other hand, the results of Cabral-Souza [33] indicate that the mechanism of action of the anti-inflammatory effect of *T. subulata* seems to be related to a decrease in the activation of the inflammatory response through MAPK, mainly through the inhibition of ERK-1/2 phosphorylation, and the decrease in the secretion of proinflammatory cytokines. So far, the three *Turnera* species with the most bioactivity reports appear to have anti-inflammatory activity. The information presented in this section corroborates the anti-inflammatory activity reported for *Turnera* spp.

### 3.3. Immunomodulatory Activity

The modulation of the immune system’s responses, either to stimulate or increase the response or, conversely, suppress it, is a therapeutic strategy that has gained interest in recent years [34].

Soromou et al. evaluated the ability of pinocembrin **(1)** (Figure 1), a flavonoid present in *T. diffusa*, to mitigate LPS-induced endotoxemia in mice. The authors found that pretreatment with 20 and 50 mg/kg of pinocembrin reduced LPS-induced mortality in mice; in addition, the 50 mg/kg dose significantly reduced levels of the proinflammatory cytokines TNFα, IL-1β, and IL-6 compared with the group treated with LPS alone [35].

Reyes et al. evaluated the phagocytic activity of *Seriola rivoliana* spleen leukocytes exposed to aqueous and methanolic extracts from the *T. diffusa* leaf. A significant increase in phagocytic activity of up to two- and three-fold was observed for the aqueous and methanolic extracts, respectively, at a 50 µg/mL concentration. They also found that the IL-1β gene was upregulated in leukocytes pretreated with the extracts and challenged with *Vibrio parahaemolyticus*. This regulation had a dose-dependent effect in a concentration range of 12.5–50 µg/mL [26].

Duarte da Luz et al. found that the aqueous and hydroalcoholic extracts of *T. subulata* flower and leaf possess anti-inflammatory activity in LPS-stimulated macrophages. In an in vitro assay with the RAW 264.7 cell line, they found that in the groups administered with the extracts, there was a significant reduction in the levels of proinflammatory cytokines such as TNF-α, IL-1β, PGE-E2, and NO compared with the untreated group. In addition, the extracts caused a significant increase in the production of IL-6, an anti-inflammatory cytokine, compared with the untreated group. The authors attribute the anti-inflammatory properties of *T. subulata* to the presence of phenolic compounds, such as vitexin **(2)** (Figure 1), since these are capable of modulating the expression of inflammatory mediators [36]. However, biodirected isolation would be an opportunity to corroborate this hypothesis.

### 3.4. Hepatoprotection

The liver is the body’s largest solid organ and fulfills many bodily functions, including hormone synthesis, toxin elimination, and energy storage. However, it is also susceptible to damage by various agents and can fall into a state of oxidative stress [37]. One of the biological activities reported for *Turnera* spp. that has dramatically increased the number of publications in the last 10 years is hepatoprotection.

Brito et al. report that oral administration for 7 and 21 days of a hydroalcoholic leaf extract of *T. ulmifolia* Linn. var *elegans* (synonym *T. subulata*) (500 mg/kg) to Wistar rats with carbon tetrachloride (CCl_4_)-induced damage significantly reduced AST and ALT levels. In addition, the histopathological analysis showed that unlike the group treated with CCl_4_ alone, which presented lobular degeneration and necrosis, the animals administered with the extract showed hepatocyte regeneration, decreased lipid deposition, and minimal centrilobular necrosis [38].

Bathula et al. administered aqueous and ethanolic extracts of *T. aphrodisiaca* Ward (Turneraceae) (synonym *T. diffusa*) leaf to rats with CCl_4_-induced damage at 200 and 400 mg/kg orally for 1 week. At the end of the study, they found a significant reduction in ALT, AST, and ALP levels in the groups treated with the extracts compared with the untreated groups. In contrast, SOD and CAT activity significantly increased [28]. Similar results were obtained with the ethanolic extract by employing an ethanol damage model and increasing the treatment period to 21 days, in addition to decreasing total bilirubin levels. In the damage group, a histopathological analysis of the liver showed cellular degeneration of hepatocytes, lymphocyte infiltrate, and loss of cellular structure, while in the groups treated with *T. aphrodisiaca*, an improvement in cellular structure was seen, with a lower amount of cellular infiltrate [39].

Hasan et al. demonstrated that *T diffusa* possesses hepatoprotective activity in a rat model of amitriptyline-induced damage. The oral administration of 80 mg/kg of an aqueous leaf extract obtained from the plant, either co-administered or administered post-treatment with amitriptyline for 28 days, significantly reduced ALT, AST, and ALP levels. In addition, they noted an improvement in the percentage of amitriptyline-induced DNA damage, evidenced by a significant reduction in DNA tail length in a comet assay in the *T. diffusa*-administered groups. The histopathological analysis revealed hepatocyte destruction, congestion with cellular infiltrate, and necrosis in the liver of amitriptyline-treated rats, whereas liver tissue from the co- and post-*T. diffusa*-administered groups showed a slight improvement in hepatocyte structure and only slight congestion [40].

Rodriguez et al. demonstrated the antifibrotic potential of a flavonoid-rich fraction obtained from an extract of the aerial part of *T. diffusa*. In this work, they observed that while LX-2 cells exposed to TGF-β, the main profibrogenic cytokine, change their morphology to an adherent spindle-like one, cells co-treated with TGF-β and *T. diffusa* maintain their stellate morphology. In addition, treatment with *T. diffusa* also decreased the expression of fibrosis markers, such as COL1α1 mRNA and α-SMA and TIMP1 proteins. The authors consider that a probable mechanism of action for the hepatoprotective activity of *T. diffusa* is through the induction of apoptosis in activated hepatic stellate cells [41].

Through biodirected fractionation, Waksman et al. found that a luteolin-based C-glycosylated flavonoid was primarily responsible for the hepatoprotective activity of *T. diffusa* [42]. They named this compound hepatodamianol **(3)** (Figure 1); this is a plant biomarker since it has only been isolated from this species. Subsequently, Delgado et al. evaluated the in vitro hepatoprotective activity of a standardized extract rich in hepatodamianol and pure hepatodamianol. Both the extract and hepatodamianol significantly reduced the AST levels of HepG2 cells with CCl_4_-induced damage. The activity of hepatodamianol was four-fold higher than silibinin in the developed model. Furthermore, they evaluated in vivo the activity of the standardized hepatodamianol-rich extract in a model with Wistar rats and the same damage inducer. The extract also significantly decreased serum AST and ALT levels compared with the untreated group; it also showed comparable activity to that of silymarin at the same dose (70 mg/kg), since there was no significant difference in AST levels between these two groups [22].

So far, *T. ulmifolia* and *T. diffusa* have shown their great hepatoprotective power both in vitro and in vivo. According to the results reviewed, the extract of the aerial part of *T. diffusa* is the most potent; a group of glycosylated C-flavonoids is responsible for this activity. The major component has been isolated and identified; it will be interesting to isolate and identify the other minority flavonoids present in the active fraction.

### 3.5. Nephroprotection

The kidney is the organ responsible for eliminating waste products from the body through urine. A sudden loss of function of this organ results in acute kidney damage, which in turn can progress to a state of chronic damage. One of the leading causes of acute kidney injury is drug-induced nephrotoxicity [43].

Hasan et al. evaluated the nephroprotective potential of *T. diffusa* in an albino rat model with kidney damage induced by amitriptyline. The administration of the aqueous extract (80 mg/kg), either in co-treatment or after administration of amitriptyline, significantly reduced the levels of urea, creatinine, Cl^−^, and Na^+^, while the K^+^ and Ca^++^ levels were increased; in addition, a comet assay showed a reduction in DNA damage and a histological analysis showed an improvement in the histological characteristics of the renal tubules compared with the group administered only with amitriptyline. Moreover, while amitriptyline treatment showed a strongly positive grade 5 reaction for p53 on immunohistochemical staining, co-treatment with *T. diffusa* decreased this positive reaction to grade 4 [44]. This is the first report we have found of the nephroprotective activity of *T. diffusa*, so a biodirected isolation study would be essential to isolate the responsible compound(s) and establish a probable mechanism of action.

### 3.6. Testicular Protection

The testicle is an organ susceptible to damage caused by oxidative stress. This damage can result from aging [45]. However, it can also be caused by exposure to substances such as pesticides, some drugs, or heavy metals. These substances cause alterations in sex hormone levels, reduction in sperm count, and disruptions in the antioxidant system, among other effects [46].

El-Demerdash et al. administered 50 mg/kg orally of a *T. diffusa* leaf extract for 3 weeks to rats exposed to fenitrothion and potassium dichromate (Cr (VI)). The individual administration of the extract, as well as its administration in conjunction with damaging agents, significantly reduced lipid peroxidation levels, increased the activity of antioxidant enzymes, reduced ALT, AST, and ALP levels, and improved testosterone and follicle-stimulating hormone (FSH) levels. In addition, they observed in histological sections that rats treated with *T. diffusa* and intoxicated with fenitrothion and Cr (VI) showed an improvement in testicular tissue architecture and sperm count, thus concluding that *T. diffusa* effectively inhibits fenitrothion- and Cr (VI)-induced testicular toxicity [29].

Similarly, Tousson et al. evaluated the potential for testicular protection conferred by *T. diffusa* in a rat model of amitriptyline-induced testicular toxicity. Eighty mg/kg of an aqueous leaf extract was administered daily, either as a co-treatment or post-treatment with amitriptyline, for 4 weeks. The administration of *T. diffusa*, alone or as an adjunct to amitriptyline treatment, improved testosterone, FSH, luteinizing hormone, and prolactin levels compared with the group treated with amitriptyline alone. Furthermore, the administration of the extract was associated with a decrease in cell damage induced by amitriptyline and an increase in the number of spermatozoa and Leydig cells. Although *T. diffusa* significantly reduced lipid peroxidation levels and increased the activity of some antioxidant enzymes (glutathione (GSH), glutathione peroxidase (GPx), glutathione reductase (GR), and SOD), there was no significant difference in glutathione S-transferase (GST) levels between the groups administered with *T. diffusa* compared with those not treated. These data suggest that *T. diffusa* represents an alternative that mitigates testicular damage induced by amitriptyline [30].

Kumar et al. evaluated both the ability to ameliorate testicular damage and to reduce the oxidative stress generated in the testis in a rat model of diabetes. Using a sperm analysis, they found that the administration of a *T. diffusa* aqueous leaf extract (100 and 200 mg/kg) for 4 weeks significantly improved the Johnsen score, blood–testicular barrier protein expression, motility, viability, and sperm count parameters compared with the untreated group. In addition, oxidative stress levels, measured through NADPH oxidase 2 and MDA, were significantly reduced with respect to the untreated group. In contrast, testicular expression levels of inflammatory markers such as NF-ΚB, p65, and TNF-α, among others, decreased compared with the untreated group [47].

The above indicates that the *T. diffusa* leaf extract can reduce testicular damage produced by oxidative stress generated by exogenous agents or conditions such as diabetes.

### 3.7. Neuroprotective Effect

Neurological diseases range from neuropsychiatric disorders to neurodegenerative diseases such as Alzheimer’s disease or Parkinson’s disease. Most of these diseases have in common oxidative stress processes, neuroinflammation, as well as neuronal death, among other aspects. In that sense, searching for neuroprotective agents that contribute to decreasing neurological damage and disease progression is vital [48].

Gomes Bezerra et al. orally administered a hydroalcoholic extract of the aerial part of *T. diffusa* (500 mg/kg) in aged Wistar rats to determine whether it exerts an anti-apoptotic effect on the hippocampus. Although the authors found no significant differences in the percentage of TUNEL-positive cells between the treated and untreated groups, they consider that this may be due to the model used since no apoptosis inducer was employed and only the hippocampal area was studied [49].

Bernardo et al. evaluated the neuroprotective potential of aqueous extracts of different plants. Among the species evaluated, *T. diffusa* presented the lowest IC_50_ values for enzyme inhibition assays, with IC_50_ of 0.352 mg/mL for acetylcholinesterase and 0.370 mg/mL for butyrylcholinesterase. Furthermore, they observed that pretreatment with the extract (125 µg/mL) significantly decreased ROS formation in SH-SY5Y cells exposed to tert-butylhydroperoxide and glutamate. On the other hand, although the extract (125 µg/mL) was not able to increase the cell viability of SH-SY5Y cells co-exposed to glutamate, it did increase the LC_50_ of glutamate from 57 to 127 mM [50]. The authors consider that the neuroprotective activity of *T. diffusa* is conferred by the high content of phenolic compounds in the plant, as these have been shown to have neuroprotective properties [51].

Bernardo et al. report that the aqueous extract of *T. diffusa* bark exhibits tyrosinase enzyme inhibitory activity, with an IC_50_ of 577 µg/mL. In addition, the extract was able to completely inhibit the activity of the enzyme lipooxygenase five at the maximum dose evaluated (385 µg/mL); the calculated IC_50_ for lipoxygenase five was 136.23 µg/mL [27]. These enzymes are closely associated with the process of neuroinflammation and the development of neurodegenerative diseases [52]. On the other hand, the extract, at doses of 50 and 200 µg/mL, could significantly and dose-dependently decrease NO production in BV-2 microglia cells stimulated with IFN-γ, with a 35 and 40% decrease, respectively, in NO levels.

In another study by Bernardo et al., nanophytosomes loaded with an aqueous *T. diffusa* leaf extract were found to significantly decrease NO production in BV-2 cells challenged with IFN-γ in a dose-dependent manner. Furthermore, they found that pretreatment of SH-SY5Y cells with the nanophytosomes protects against glutamate-induced toxicity by significantly increasing the percentage of cell viability compared with the untreated group, but does not protect against 6-hydroxydopamine/ascorbic acid-induced toxicity [53].

Velez et al. reported that pretreatment with ethanolic, chloroform, and hexane extracts of *T. diffusa* leaf increases the cell viability of PC-12 cells exposed to glutamate compared with cells pretreated with ascorbic acid. However, in the evaluation of caspase-3 activity, no significant difference was found in the group pretreated with the ethanolic extract compared with the group without pretreatment or damage, so the protective mechanism of the extract must be different from the inhibition of apoptosis [23].

In summary, Bernardo et al. have demonstrated the potent neuroprotective activity of the aqueous extract of *T. diffusa* leaves, while Velez et al. showed activity of less polar extracts. It would be interesting to further investigate the mode of action of each extract.

### 3.8. Hypoglycemic/Antidiabetic/Antihyperglycemic Activity

Diabetes mellitus is a disease characterized by high blood glucose levels, either because of an inability to produce insulin or because the body is resistant to its effect. It is a global public health problem since it is estimated that by 2040, 642 million people will suffer from this disease [54].

The antidiabetic effect of extracts obtained from *Turnera* spp. using animal models has already been identified by Szewczyk et al. [4]. Additional evidence has emerged in recent years reinforcing the use of *Turnera* spp. to lower glucose levels. Parra et al. found that the methanolic extract of the leaf and stem of *T. diffusa* exerts a hypoglycemic effect in normoglycemic mice. By biodirected isolation, the terpenoid teuhetenone A **(4)** (Figure 1) was identified as the primary agent responsible for the activity. This compound, administered intraperitoneally at 1 and 5 mg/kg doses, generated a 40% decrease in glucose levels in normoglycemic mice. In mice with alloxan-induced diabetes, there was an effect only with the 5 mg/kg dose, with a sustained reduction in glucose levels of about 30% during the 6 h of the study. Teuhetenone A was not able to inhibit the activity of the α-glucosidase enzyme, so the hypoglycemic and/or antidiabetic activity of the compound probably follows a mechanism of action other than the inhibition of intestinal carbohydrate absorption [55].

According to Esquivel et al., the oral administration of 150 mg/kg of the hydroalcoholic extract of the aerial part of *T. diffusa* for 5 weeks did not modify blood glucose levels in rats with streptozotocin-induced diabetes compared to the untreated group [16]. These authors obtained similar results when extending the administration period to 8 weeks since neither the hydroalcoholic extract of *T. diffusa* var. *diffusa* nor *T. diffusa* var. *aphrodisiaca* caused a significant decrease in glycemia in the treated groups [56].

Adame et al. evaluated the antihyperglycemic activity of several plant extracts in different models. Among the extracts evaluated, the methanolic extract of *T. diffusa* did not show inhibitory activity of α-glucosidase or α-amylase enzymes; nor did it affect intestinal glucose absorption in the inverted sac test, as the extract did not generate a significant reduction in glucose area under the curve (AUC) with respect to the negative control. Conversely, in the glucose tolerance test with rats, there was a significant reduction of glucose AUC with respect to the starch control at doses of 0.5–5 mg/kg [57].

Rebouças et al. report that a 4-day administration of an ethanolic extract of *T. subulata* flower (4–40 mg/kg) significantly reduced glucose levels in hyperglycemic *D. rerio*, commonly known as zebrafish [24].

Kumar et al. administered an aqueous extract of *T. diffusa* leaf (100 and 200 mg/kg/day) to rats with streptozotocin-induced diabetes for 8 weeks. In their work, fasting glucose levels were monitored weekly and were significantly lower in the treated groups than in the untreated group. In addition, they found that plasma insulin levels at the end of treatment were significantly higher in the groups treated with the extract compared with the untreated group [47].

Despite the differences in the results reported by Esquivel [16,56] concerning the rest of the publications discussed in this section, this could be due to methodological differences between the studies, since both the dose and the route of administration of the extracts were different. In addition, the results of Kumar et al. [47] hint at a possible mechanism of action through increased insulin secretion. In conclusion, the additional information regarding the ability of *Turnera* spp. to reduce glucose levels in the different models evaluated reinforces the use of species of this genus as a hypoglycemic/antidiabetic and/or antihyperglycemic agent.

### 3.9. Anti-Obesity Activity

Obesity is a growing health problem because of its high prevalence and the risk factor it represents for the development of other chronic degenerative diseases. Treatment alternatives for obesity include lifestyle and dietary modifications, surgical procedures, and the use of drugs and medicinal plants [58].

Most of the studies that have evaluated the anti-obesity activity of *T. diffusa* used YGD, which is a typical South American herbal preparation that contains yerba mate (*Ilex paraguariensis* A.St.-Hil. (Aquifoliaceae)), guarana (*Paullinia cupana* Kunth (Sapindaceae)), and damiana (*T. diffusa*). In their review, Szewczyk et al. [4] presented some clinical studies carried out with YGD. This preparation was able to increase gastric emptying time while reducing body weight, food intake, kilocalories, and energy, as well as favoring the feeling of satiety.

Ruxton et al. reported similar results. They conducted an uncontrolled study in healthy overweight adults consuming Zotrim^®^ for at least 6 weeks. Zotrim is a proprietary herbal preparation prepared from YGD and is commercially available. The authors found that at the end of 10 weeks, the subjects presented a significant reduction compared with the beginning of the study in the parameters of weight, body mass index, and hip and waist circumference. On the other hand, at the end of 6 weeks, there were also significant changes regarding the sensations of hunger and satiety [59].

Moreover, Dorantes et al. found that the daily administration for 28 days of 100 and 1000 mg/kg of an aqueous extract of *T. diffusa* leaves to mice significantly decreased body weight gain [60].

Most studies suggesting that *T diffusa* possesses anti-obesity activity were done with the YGD mixture. However, since it is a mixture of plants, it is impossible to determine whether this activity is given by only one of the mixture’s components or by a synergism between them. Additional studies evaluating only *T. diffusa* are needed to corroborate its anti-obesity effect.

### 3.10. Analgesic Activity

According to the International Association for the Study of Pain, *pain* is defined as “an unpleasant sensory and emotional experience associated with actual or potential tissue damage or described in terms of such damage”. Pain can be acute or nociceptive and chronic and in turn, can be classified as neuropathic or inflammatory [61].

According to Antonio et al., the hydroalcoholic extract of *T. ulmifolia*, as well as its ethanolic fraction, do not have an analgesic effect in the writhing test induced by acetic acid in mice [62].

Kumar et al. evaluated the analgesic effect of apigenin **(5)** (Figure 1), a flavone isolated from the methanolic extract of the aerial part of *T. aphrodisiaca*, using the tail dip test. Doses of 2, 5, and 10 mg/kg were administered orally. The authors reported that apigenin had a dose-dependent analgesic effect comparable to morphine, which was used as a positive control. The greatest effect was observed 30 min after the administration of 10 mg/kg apigenin, and was maintained up to 4 h post-administration [63].

Lawal et al. studied the antinociceptive activity of EO from the leaves and stem bark of *T. diffusa* using the hot plate model in rats. Both the EO obtained from the leaves and that obtained from the stem bark showed an antinociceptive effect at doses of 100–400 mg/kg administered orally, evidenced by a significant increase in the reaction time of the rats compared with the control; this effect lasted during the 120-min testing [32].

Rebouças et al. reported that the hydroalcoholic extract of *T. subulata* flowers administered intramuscularly in doses of 4–40 mg/kg can reverse the nociceptive behavior induced by formalin in an animal model with *D. rerio*. Through an HPLC-DAD analysis, they identified chlorogenic acid **(6)** (Figure 1), apigenin, luteolin-7-O-glucoside **(7)** (Figure 1), and vitexin as the major components in this extract. The analgesic activity of the extract is exerted through the ankyrin transient potential receptor 1 (TRPA1) since, when administering camphor—an antagonist that inhibits the analgesic effect—the effect of the *T. subulata* extract is blocked. Moreover, molecular docking studies showed that these four compounds have a high affinity for TRPA1 [24].

Two of the three species reported in the analgesic activity assays produced in vivo effects; the report by Reboucas et al. [24] provides evidence of a possible mechanism of action. Although chlorogenic acid and some flavonoids have been identified in the active extract, it is essential to continue isolation studies to establish which of the components of the extract are responsible for this activity.

### 3.11. Antidepressant and Anxiolytic Activity

Depression and anxiety are mental illnesses with a high prevalence worldwide that usually co-occur in people who suffer from them. The World Health Organization reports that 322 million people worldwide suffer from depression, while anxiety affects 264 million. These diseases are a public health problem since they contribute to a large extent to disability in the world [64].

In their review, Szewczyk et al. [4] described the anxiolytic activity of extracts and compounds obtained from *T. diffusa*. In our new search, we found an additional publication that reinforces the use of *T. diffusa* to treat mental disorders such as depression and anxiety. Dorantes et al. explored the anxiolytic effect of an aqueous extract of *T. diffusa* leaf using the elevated maze model and the antidepressant potential with the forced swim model in male mice. The acute administration of a single dose of the extract showed anxiolytic activity with a dose-dependent behavior (10–200 mg/kg), expressed as increased residence time and entries in the open arm of the maze. On the other hand, the time spent on the central platform was also significantly increased, but only at the lowest dose (1 mg/kg), and this effect decreased as the dose increased. Regarding the antidepressant activity, a single dose of the extract did not show an effect, while the administration of three doses in a 24-h period (200 mg/kg) significantly reduced the immobilization time in the forced swimming test. This effect was comparable to that of the positive control of fluoxetine (10 mg/kg) [60].

### 3.12. Aphrodisiac Activity

According to Sandroni, aphrodisiacs are substances capable of awakening the sexual instinct through increased sexual desire, potency, or sexual pleasure. These substances exert their effect at the central nervous system level by altering the levels of some neurotransmitters or sex hormones [65].

Szewczyk et al. [4] has already described some of the trials in animal models reported to evaluate the aphrodisiac activity of *T. diffusa*. In addition, there are reports of clinical trials that have evaluated the ability of *T. diffusa* to exert an aphrodisiac effect, administered mainly as a supplement in a mixture with other plants.

Shah et al. conducted a clinical study evaluating the safety and effectiveness of an herbal mixture marketed as VigRx Plus, which contains *T. diffusa* among its ingredients, to manage erectile dysfunction. They found that, at the end of 12 weeks, there was a significant increase in the International Index of Erectile Dysfunction score in the treated group compared with the placebo group, without causing changes in sperm parameters or testosterone levels [66].

Palacios et al. conducted a pilot study to evaluate the efficacy of the Libicare^®^ supplement on sexual function in postmenopausal women. This supplement consists of a mixture of four herbs, including *T. diffusa*. The administration of two daily tablets of the supplement for 9 weeks significantly improved five of the six parameters evaluated at the end of the study. In addition, 86% of patients improved their Female Index of Sexual Function score. There was a significant increase of 0.09 units in the overall testosterone level and a significant decrease in the overall level of sex hormone-binding globulin between the beginning and the end of the study. The authors associate the increase in lubrication and orgasm parameters with the anti-aromatase activity reported for *T. diffusa* [67].

Kuchernig observed that the overall effect of the combined individual effects observed for ingredients of the plant damiana (*T. diffusa*) could be a promising treatment option with a positive influence on decreased libido. The different mechanisms of action give good evidence that drugs made from damiana can positively influence decreased libido. Apigenin is an efficacy-determining component of anxiolytic and analgesic activity. Both activities can play an important role, especially the anxiolytic effects that can help clear the head. In addition, the total extract leads to smooth muscle relaxation and thus, presumably via the NO-cGMP pathway, to a blood flow enhancing effect, which may make its contribution via intensification of arousal. Furthermore, the aromatase inhibitory effect of the total extract may enhance desire by increasing free testosterone. The authors conclude that as a phytotherapeutically approved drug in the appropriate dosage (e.g., Remisens^®^, one film-coated tablet three times a day), a therapy trial may be helpful or complement other measures as a building block of the therapy [68].

Similarly, Rotmann et al. conducted a study to gain further insights into the effects and tolerability of a traditional herbal medicinal product made with *T. diffusa* in the treatment of women suffering from female sexual interest/arousal disorder (FSIAD). The patients (*n* = 35, 46.1 ± 10.9 years) showed a significant increase in their Female Sexual Function Index. The score of the Female Sexual Distress Scale–Revised decreased significantly, while the quality of life (measured by Münchner Lebensqualitäts-Dimensionen Liste) tended to increase. This study showed that in patients with FSIAD, *T. diffusa* may improve individual symptoms. These results are of great importance, considering that damiana leaf extracts are now the only pharmaceutical therapy option in Germany for treating women suffering from FSIAD [69].

The aphrodisiac activity of *T. diffusa* is one of the most reported bioactivities for this species. Although some of the clinical trials performed evaluated *T. diffusa* in mixtures, the results obtained could be attributed, at least in part, to the presence of *T. diffusa* in the mixture since the aphrodisiac activity of this species alone has already been demonstrated in animal models and in at least one clinical trial (Rotmann [69]). The evidence shown above supports the use of *T. diffusa* as an aphrodisiac.

### 3.13. Herbal–Drug Interactions

Humans have specialized enzymes and transporters that modulate the absorption, metabolism, distribution, and excretion of drugs. In turn, the activity of these enzymes and transporters can be affected by other substances administered simultaneously with the drugs, thus generating interactions that can affect drug efficacy and toxicity. When these interactions are caused by phytochemicals, we speak of drug–herb interactions. These interactions can affect the pharmacodynamic parameters of the drugs or be of a pharmacokinetic type, where the activity of enzymes or transporters involved in the metabolism of the drug is modified [70].

Husain et al. screened 123 hydroalcoholic plant extracts to analyze their effect on nuclear receptors and enzymes involved in the metabolism of drugs and xenobiotics and to thus find possible drug–herb interactions. Of the extracts studied, the *T. diffusa* leaf extract was a potent activator of the pregnane-xenobiotic (PXR) and aryl-hydrocarbon (AhR) receptors, with ≥four- and ≥ten-fold increases in transcriptional activity, respectively, whereas the extract of a mixture of different parts of the plant only presented a two- to <three-fold increase for PXR and from >two- to <five-fold increase for AhR. The leaf extract was also found to be a potent inhibitor of CYP3A4 activity, with IC_50_ around 6 µg/mL, while that of the mixed parts was about 12.5 µg/mL. Likewise, both extracts also had CYP1A2 enzyme inhibitory activity, with IC_50_ values of around 14 µg/mL for the leaf extract and 24 µg/mL for the plant extract mixture [71]. These findings demonstrate the interaction of *T. diffusa* leaf extracts on drug metabolism, perhaps related to its hepatoprotective effect, since CYP3A4 and CYP1A2 are localized in the liver and intestine.

### 3.14. Anti-Glycosylation Activity

Advanced glycosylation end-products (AGEs) are molecules that can be formed exogenously or endogenously through nonenzymatic reactions. These reactions occur between a nonreducing sugar such as glucose and other macromolecules such as nucleic acids, proteins, or lipids. AGEs can promote a state of oxidative stress and inflammation. In addition, the increase in their levels is associated with the development and progression of different chronic degenerative diseases [72].

Bernardo et al. studied the potential to inhibit the production of AGEs of an aqueous extract of *T. diffusa* bark. They found that the extract in concentrations of 8–133 µg/mL can inhibit the formation of AGEs from bovine serum albumin over 14 days. The authors attribute this activity to the high content of flavonoids present in *T. diffusa* since these can inhibit AGEs formation [27,73].

Viel et al. demonstrated the anti-glycosylation activity of the hydroalcoholic extract of the aerial part of *T. ulmifolia* in microcapsules (10 mg/mL) since it significantly reduced the percentage of free amino groups with respect to aminoguanidine (AMG), used as a positive control, and inhibited the formation of AGEs in a comparable way to AMG. Furthermore, in the analysis of the electrophoretic profile of bovine serum albumin (BSA), the lane of BSA with the *T. ulmifolia* extract showed a similar electrophoretic profile to that of BSA with AMG, thus demonstrating that the extract prevents BSA glycosylation in a similar way to AMG [25].

### 3.15. Inhibitory Activity of Monoamine Oxidases

Monoamine oxidases (MAOs) are a family of enzymes that catalyze deamination reactions of amines and neurotransmitters. The inhibition of these enzymes has been used as a therapeutic target for treating different neurodegenerative and psychiatric diseases [74].

Bernardo et al. evaluated the ability of different plant extracts to inhibit MAO-A. Of the plants evaluated, the aqueous extract of *T. diffusa* leaf showed the most significant potential as an inhibitor of MAO-A, with an IC_50_ of 129.80 µg/mL [50].

Chaurasiya et al. isolated a series of flavonoids and glycosylated flavonoids from *T. diffusa*, which had inhibitory activity on MAO-A and -MAO-B to different degrees. Among them, acacetin **(8)** (Figure 1), acacetin 7-methyl ether **(9)** (Figure 1)**,** and velutin **(10)** (Figure 1) showed selective inhibition towards MAO-B. In particular, acacetin 7-methyl ether had a >500-fold higher selectivity for MAO-B. Enzyme kinetic studies showed that the inhibition of acacetin and acacetin 7-methyl ether on MAO-B is of a competitive type, with inhibition constants of 36 and 45 nM, respectively; furthermore, the binding of acacetin 7-methyl ether with MAO-B is partially reversible, since upon the analysis of the enzyme activity after a dialysis process, only about 80% of its activity was recovered. Docking and molecular dynamics studies confirmed that there is a better binding affinity for MAO-B compared with MAO-A, and that acacetin 7-methyl ether presents strong hydrophobic interactions with Ile199 and Ile316 residues, which are critical for selectivity towards MAO-B [75]. This inhibition is possibly related to the antidepressant activity reported by Dorantes et al. [60], but it also raises the question of a potential beneficial effect of *T. diffusa* in Parkinson’s disease and other neurodegenerative diseases related to dopamine availability.

### 3.16. Inhibitory Activity of Phosphodiesterase 5 (PDE-5)

Phosphodiesterases (PDEs) are a family of enzymes responsible for cleaving the 3′-ribose-phosphate bond of cyclic adenosine monophosphate and cyclic guanosine monophosphate. In particular, the inhibition of the PDE-5 isoform is relevant in treating cardiovascular diseases and erectile dysfunction, among other conditions [76].

Feistel et al. evaluated the inhibitory activity of PDE-5 of several species and varieties of *Turnera* in an in vitro model. The authors report that the crude hydroalcoholic extract of *T. ulmifolia* and *T. diffusa* have an IC_50_ of 10–20 µg/mL. Furthermore, the authors report that a semi-purified fraction shows an IC_50_ of 5 µg/mL [77].

### 3.17. Anti-Aromatase and Estrogenic Activity

Aromatase is an enzyme involved in the conversion of androgen precursors into estrogens. The use of inhibitors of this enzyme has been explored for the treatment of breast cancer, gynecomastia, and infertility, among other diseases [78,79].

Zava et al. determined the binding capacity of plant extracts on estrogen and progesterone receptors in breast cancer cell lines. Among the species evaluated, the hydroalcoholic extract of *T. diffusa* showed binding to the progesterone receptor; however, in the alkaline phosphatase induction assay, it was found that damiana extract neither induced nor inhibited this enzyme, which is why it is considered to have neutral activity on the progesterone receptor [80].

The anti-aromatase and estrogenic activity of *Turnera* has been previously explored in the review carried out by Szewczyk et al. [4], where it was reported that the secondary metabolites pinocembrin and acacetin isolated from *T. diffusa* have inhibitory activity of the aromatase enzyme, while apigenin 7-glucoside **(11)** (Figure 1), Z-echinacin **(12)** (Figure 1), and pinocembrin have estrogenic activity. These results are further supported by the findings of Powers et al., who conducted a molecular docking study to identify possible compounds with estrogenic activity present in herbal supplements. The authors reported that two compounds present in *T. aphrodisiaca* exhibited coupling with estrogen receptors. Pinocembrin showed interactions of −81.4 and −87.9 kJ/mol for estrogen receptors Erα and Erβ, respectively, while luteolin-8-propenoic acid **(13)** (Figure 1) had values of −113.1 and −123.1 kJ/mol, respectively [81].

### 3.18. Anti-Photoaging Activity

Exposure to UV radiation generates alterations in the skin, resulting in photoaging. This process can increase oxidative stress levels, as well as induce cell damage at the level of DNA and the extracellular matrix and generate a state of immunosuppression [82].

Kim et al. demonstrated the anti-photoaging potential of a hydroalcoholic extract from the *T. diffusa* leaf in keratinocyte (HaCaT) and human dermal fibroblast (HDF) cell lines exposed to UVB radiation. They found that at a concentration of 50 µg/mL, the extract significantly reduced the levels of ROS and enhanced the expression levels of regulators and antioxidant proteins such as dihydrolipoamide dehydrogenase and heme-oxygenase 1 in both cell lines. Upregulation of metalloproteinase type I (MMP-1) is a hallmark of photoaging; *T. diffusa* extract decreased the expression and secretion of MMP-1 and increased that of procollagen type I. It also promoted the activation of TGF-β1, which acts on the expression of collagen, and decreased the levels of phosphorylated members of the mitogen-activated protein kinase (MAPK) pathway. With these results, the authors concluded that the anti-photoaging activity of the extract is exerted through the regulation of the MMP-1/procollagen type 1, TGF-β1/Smad, and MAPK pathways [20].

### 3.19. Skin Penetration Enhancement Activity

Among the routes of drug administration is the transdermal route. However, for adequate delivery of the active ingredient, a drug must pass through the stratum corneum layer of the skin, which is highly selective. One of the strategies employed to achieve this delivery is the use of chemical penetration enhancers. These are substances that, through various mechanisms, reversibly modify the skin’s permeability, thereby increasing the entry of substances through this route. Fatty acids, essential oils, and surfactants are just a few examples of such compounds [83].

Carreño et al. evaluated the ability of different EOs and some constituents present therein to increase caffeine penetration into the skin in an in vitro assay using Franz cells and mouse skin. For this purpose, the authors prepared hydrogels based on Carbopol 934, which contained caffeine as a model molecule and the EO or compound. Among the EOs evaluated, *T. diffusa* significantly increased the caffeine permeation coefficient and steady-state flux, with an enhancement index of about 3. In contrast, the hydrogel with β-caryophyllene **(14)** (Figure 1), a component in the EO of *T. diffusa*, had an enhancement index of 1.23, an increase that was insignificant compared to the hydrogel containing caffeine alone. Additionally, the authors demonstrated that the 21-day administration of the prepared hydrogels (hydrogel + caffeine + EO) did not cause irritation or damage to the skin of mice [84].

### 3.20. Anti-Angiogenic Activity

Angiogenesis is the proliferation and migration of endothelial cells from pre-existing blood vessels, with the consequent formation of new blood vessels. This process is fundamental in tumor development, since blood vessels provide oxygen and nutrients to cancer cells, which allows the tumor to develop and metastasize [85]. For this reason, one of the treatment alternatives for cancer is based on the search for and development of molecules with anti-angiogenic activity.

Saravanan et al. studied the anti-angiogenic potential of *T. subulata* extracts in a chicken chorioallantoic membrane model. Their results showed that the chloroform, ethyl acetate, and ethanol extracts obtained from the whole plant at doses of 50 and 100 µg/mL significantly reduced the size of the pre-existing vasculature in 24 h [21]. These findings demonstrate a potential antiangiogenic activity of *T. subulata*, so it is interesting to continue evaluating this activity with other species of the genus and isolate and identify the compounds responsible for this action.

### 3.21. Anti-Coagulant Activity

Coagulation is a complex process involving many protein factors and is vital for proper circulation and the maintenance of hemostasis. Alterations in the coagulation process contribute to the development of cardiovascular and thromboembolic diseases, which constitute a severe health problem worldwide [86].

Duarte da Luz et al. evaluated the anti-coagulant potential of *T. subulata* leaf extracts. They found that both the hydroalcoholic extract and, to a greater extent, the ethyl acetate fraction of *T. subulata*, inhibited clot formation. This effect is exerted through inhibiting thrombin action. In addition, *T. subulata* presented a lower hemorrhagic rate than heparin. The authors consider that the anti-coagulant action is due to the high amount of glycosylated flavonoids present in *T. subulata*, as these have been reported to have anti-coagulant activity [86,87].

### 3.22. Smooth Muscle Relaxant Activity

Hnatyszyn et al. evaluated the effect of extracts from Argentine plants on the relaxation of the smooth muscle of the corpus cavernosum of guinea pigs and compared them against *T. diffusa*. Dichloromethane and methanol extracts of *T. diffusa* caused 89% and 86% relaxation at 10 mg/mL and 58% and 62% relaxation at 5.0 mg/mL, respectively [88]. In their review, Szewczyk et al. [4] referenced these results, although they classified this activity as antispasmodic rather than as smooth muscle relaxant. Although antispasmodics exert their effect through smooth muscle relaxation, this term is more frequently used to refer to drugs used for the treatment of abdominal pain [89].

On the other hand, Aguirre et al. reported that a methanolic extract of *T. diffusa* leaf does not exert a significant vasodilator effect in an ex vivo model consisting of rat aortic rings precontracted with norepinephrine, obtaining EC_50_ values >500 µg/mL and an E_max_ of 31.4% [19].

Although the results reported above yielded apparently contradictory results, this could be due to differences in the concentration used for the assays, as Hnatyszyn et al. [88] used a dose of *T. diffusa* that was 10 times higher than that used by Aguirre et al. [19]. It is possible that both results are congruent, and a large amount of extract is necessary to produce muscle relaxation.

### 3.23. Antiparasitic Activity

Garcia et al. screened 21 plants to determine their activity against the parasite *Leishmania amazonensis*. Among the plants evaluated, *T. ulmifolia* at a concentration of 100 µg/mL inhibited the growth of the parasite in the promastigote stage by 17.8%. This activity was not considered important because it did not present an inhibition of at least 50% [90].

Santos et al. evaluated the antiparasitic activity of an ethanolic extract of *T. ulmifolia* leaf against *Trypanosoma cruzi* and *Leishmania brasiliensis*. At the maximum dose evaluated (500 µg/mL), the extract presented a percentage of antiepimastigote activity of 29% for *T. cruzi* and a percentage of antipromastigote activity of 9% for *L. brasiliensis* [91].

Oliveira et al. tested the antihelmintic activity of plant extracts from the Brazilian savannah against *Haemonchus contortus*. They found that the hydroalcoholic and hydro-acetone extracts of *T. ulmifolia* root and leaf exhibit anthelmintic activity against the larval development stage of *H. contortus*. Furthermore, this activity is due, at least in part, to the presence of polyphenolic compounds in the extract. Upon an addition of polyvinylpolypyrrolidone, a compound that binds to polyphenols, the anthelmintic activity was affected, with a significant reduction in the percentage of non-hatching larvae and an increase in the percentage of stage 1 larvae present in the sample [92].

### 3.24. Antiviral Activity

Silva et al. analyzed the influence of the composition of the EOs of different plants on the antiviral activity against two serotypes of dengue virus. In this study, two samples of *T. diffusa* EO were analyzed; it was found that only the EO containing alcoholic and phenolic monoterpenes, as well as a higher percentage of hydrocarbon sesquiterpenes, presented antiviral activity, with an IC_50_ of 29–54 µg/mL and selectivity index of 7.7–14.3. Among the components present in the active EO were carvacrol **(15)** (Figure 1) and δ-cadinene **(16)** (Figure 1), which showed interactions with some of the virus proteins in molecular coupling studies [93].

### 3.25. Antifungal Activity

The antifungal activity of *T. ulmifolia* against *Candida* spp. has already been addressed by Szewczyk et al. [4] in their last review about *Turnera*. Further investigations showed that *T. subulata* and *T. diffusa* also have activity against *Candida* spp. Table 6 summarizes the results regarding the antifungal activity of *Turnera* spp. against *Candida* spp. and other fungi.

According to Kalimuthu et al., the ethanolic extract from callus of *T. ulmifolia*, at 60 µg/mL, presented growth inhibition zones (IZ) of 8 and 6 mm for *Candida albicans* and *Trichoderma viride*, respectively. These values were close to those of streptomycin, which was used as a positive control [14].

According to Saravanan et al., chloroform, ethyl acetate, and ethanol extracts of *T. subulata* possess good antifungal activity against *Aspergillus niger* and *Candida albicans*, with an IZ from 9 to 12 mm for both fungi [97].

Baez et al. evaluated the susceptibility of *C. albicans* to the antifungal activity of methanolic and hexanic leaf extracts of *T. diffusa* var. *diffusa* and *T. diffusa* var. *aphrodisiaca*. Only the hexanic extract of both varieties showed activity, with an IZ of approximately 12 mm. However, these results were statistically inferior to the positive control nystatin [95].

Tapia et al. evaluated the antifungal activity of methanolic extracts of the leaf and stem of *T. diffusa* by inhibition assays of the enzyme endo 1,3-β glucanase, involved in the synthesis of the fungal wall in some fungi, and inhibition of *Botrytis cinerea* spores. By biodirected isolation, they showed that luteolin **(17)** (Figure 1) and apigenin, at a concentration of 25 mM, inhibited the activity of endo 1,3-β glucanase by 60% and 89%, respectively, and that these two compounds exert a synergistic effect since, when evaluated together, they completely inhibited the activity of the enzyme. Furthermore, at a concentration of 100 mM, apigenin can completely inhibit the germination of *B. cinerea* spores, while luteolin does so by 90% at the same concentration [98].

Ong et al. evaluated the antifungal activity of six extracts of *T. subulata* on different fungal species. Table 6 shows the minimum inhibitory concentration (MIC) values obtained. Notably, the extracts with the lowest polarity showed the best activity [96].

Sri et al. reported that the ethanolic extract of the *T. subulata* flower presented a growth IZ of 12 mm against *C. albicans* and *A. niger*; however, this value was much lower than that of the nystatin positive control (35 mm) [99].

Dilkin et al. reported that *T. ulmifolia* presents inhibition percentages of 7.50% and 17.50% against *Rhizoctonia solani* and *Macrophomina phaseolina*, respectively [100].

Domfeh et al. evaluated the activity of herbal mixtures on the growth of a clinical isolate of *C. albicans* by the disk diffusion method. Among the mixtures tested, only the one consisting of *Centella. asiatica* (L.) Urb. (Apiaceae) sap and leaves of *Turnera microphylla* Desv. (Turneraceae) (synonym *T. diffusa*) and *Vitex agnus-castus* L. (Lamiaceae) showed activity with an inhibition halo of 19 mm. The authors argue that although *C. asiatica* and *V. agnus-castus* do have reports of antifungal activity, this is not the case for *T. microphylla*, so it would be necessary to conduct additional studies to corroborate that the activity is due to a synergistic effect between the three species that make up the mixture [101]. However, this assertion could be due to the fact that *T. microphylla* is a poorly known synonym for *T. diffusa*. Based on reports of antifungal activity of *T. diffusa*, it is likely that the observed antifungal activity of the mixture is due, at least in part, to the presence of *T. diffusa* in it.

On the other hand, Andrade et al. evaluated the fungicidal activity of the ethanolic extract of *T. subulata* leaf against *Candida* spp. They found that the extract does not have significant activity since an IC_50_ of 7544 µg/mL was obtained for *C. albicans*, while for *Candida krusei* it was 16,087.37 µg/mL. Similarly, the extract could only inhibit hyphal formation *in C. albicans* and *Candida tropicalis* at high concentrations [102].

As reported by Oliveira et al., the antimicrobial activity of plant extracts can be considered as highly active when the MIC is <100 µg/mL; when the MIC ranges from 100 to 500 µg/mL they are classified as active. MICs between 500–1000 µg/mL and 1000–2000 µg/mL are categorized as moderately and low active, respectively. An MIC of >2000 µg/mL is considered not active [103]. According to this classification, most of *Turnera* extracts possess antifungal activity; the extracts of *T. diffusa* var. *aphrodisiaca* and *T. diffusa* var. *diffusa* showed high activity on *C. albicans*; three extracts of *T. subulata* were highly active on *Cryptococcus neoformans*. Notably, *Candida parapsilosis* was susceptible to all tested extracts except for the aqueous extract. In addition, it was the only strain that presented resistance towards the antifungal activity of *T. subulata*.

### 3.26. Antibacterial Activity

Reports on the antibacterial activity of *Turnera* spp. have already been explored in a previous review on this genus [4]. Since then, at least twelve additional publications concerning the evaluation of the antibacterial activity of plants of the genus *Turnera* have been identified. Details of these reports are given in Table 7.

Arantes et al. evaluated the susceptibility of *Mycobacterium fortuitum* to polar and non-polar extracts of various plants, including *T. ulmifolia*. Polar extracts showed moderate activity, with an MIC of >500 µg/mL. However, the MIC of non-polar extracts showed promising antimycobacterial activity, with an MIC of 125 µg/mL [108].

Coutinho et al. evaluated the ability of an ethanolic leaf extract of *T. ulmifolia* to modify the resistance to aminoglycosides of resistant strains of *Escherichia coli* [109]. The extract, at a concentration of 128 µg/mL, modulated the susceptibility of *E. coli* resistant to aminoglycosides by decreasing the MIC of amikacin, neomycin, and tobramycin to half of its value.

Snowden et al. indicated that the ethanolic extract of the *T. diffusa* leaf has moderate bactericidal activity against *Staphylococcus aureus*, with a reduction of 10^2^–10^3^ in the bacterial concentration [105].

Silva et al. determined the antibacterial and antibiofilm activity of different species from the Brazilian Caatinga area. Among the species evaluated, three species of *Turnera* were evaluated: *Turnera hermannioides* Cambess (Turneraceae), *Turnera melochioides* Cambess (Turneraceae), and *T. subulata*. Of these, only the aqueous extract of the *T. melochioides* leaf has antibiofilm activity, by allowing only 26% of biofilm formation compared with the control without treatment [110].

Kalimuthu et al. tested the antibacterial activity of ethanolic and methanolic extracts of *T. ulmifolia* callus on clinical isolates of *Streptococcus. pyogenes*, *S. aureus*, *E. coli*, and *Klebsiella pneumoniae*. They found that the extracts exhibited a greater inhibition halo for all concentrations tested (20–60 µg/mL) than the streptomycin control [14]. At the highest concentration tested, the ethanolic extract had a greater effect on Gram-positive bacteria *S. aureus*, with an IZ of 18 mm, while for the methanolic extract the largest IZ was observed for Gram-negative bacteria *K. pneumoniae*, with an inhibition halo of 22 mm.

Saravanan et al. reported that the chloroform and ethanol extracts of *T. subulata* show good antibacterial activity against *E. coli* and *S. aureus*, while the ethyl acetate extract has only weak activity against these same bacteria. Although the extracts showed inhibition halos between 7 and 10 mm, these were not greater than that of the positive control ampicillin (13–15 mm) [97].

Baez et al. tested the antibacterial activity of hexanic and methanolic leaf extracts of *T. diffusa* var. *diffusa* and *T. diffusa* var. *aphrodisiaca* against urinary pathogens. The extracts of both varieties showed similar activity against *Enterococcus faecalis* and *K. pneomoniae* (IZ approximately 10 mm). The hexanic extract of *T. diffusa* var. *diffusa* and the methanolic extract of *T. diffusa* var. *diffusa* and *T. diffusa* var. *aphrodisiaca* had activity against *S. aureus* (IZ 10.34–14.89 mm), whereas *E. coli* was only susceptible to the hexanic extract of *T. diffusa* var. *diffusa* (IZ approximately 10 mm) [95].

Reyes et al. tested the antibacterial activity of the aqueous and methanolic extracts of *T. diffusa* leaf against *V. parahaemolyticus*. They found that the extracts, at doses of 200–800 µg/mL, reduced bacterial growth approximately 50% relative to the control [26].

According to Ong et al., an ethanolic extract of the aerial part of *T. subulata* has bactericidal activity against various Gram-positive and Gram-negative bacteria [96].

Freitas et al. found that the methanolic extract of the *T. subulata* leaf, in addition to possessing antibacterial activity, is capable of modulating the susceptibility of *Pseudomonas aeruginosa* to the antibiotics amikacin and gentamicin, significantly reducing the MIC, while *S. aureus* and *E. coli* did not show the same response [104].

Barrios et al. report that *T. diffusa* EO has low antibacterial activity, against both the methicillin-sensitive and the methicillin-resistant strains of *S. aureus*. In addition, the EO (200 µg/mL) is capable of significantly increasing the sensitivity to penicillin G, methicillin, and ampicillin antibiotics by decreasing the MIC by 8, 16, and 263 times, respectively [106].

Sri et al. tested the antibacterial activity of the ethanolic extract from the leaves of *T. subulata*. The extract produced an IZ of 14 mm for *E. coli* and 18 mm for *S. aureus* [99].

According to the classification proposed by Oliveira [103], most of the extracts of *Turnera* spp. present a moderate antibacterial activity, with an MIC from 500 to 1000 µg/mL. Notably, the antibacterial activity of *Turnera* spp. seems to have a broad spectrum, since Gram-positive, Gram-negative, and mycobacteria were susceptible to its antibacterial effect. The most promising results were obtained with extracts of *T. diffusa* var. *aphrodisiaca* and *T. diffusa* var. *diffusa* on wild-type ATCC strains of *K. pneumoniae* and *E. coli*, with MIC values of 50 µg/mL.

### 3.27. Anti-Biofouling Activity

Biofouling is the process of colonization by microorganisms, plants, algae, or animals on submerged surfaces, resulting in biofilm formation. This process represents a significant economic burden as it causes the corrosion of these structures, which generates costs and monetary losses [111].

Agostini et al. explored the antifouling potential of the aqueous extract of different plants from Brazil’s Caatinga region. Among them, the extract of *T. hermannioides* leaves at concentrations of 0.5 mg/mL and 1.0 mg/mL reduced the bacterial density present in a biofilm composed of proteobacteria and Bacteroidetes by more than 80%, despite not presenting antibacterial activity [112].

### 3.28. Insecticidal–Larvicidal Activity

Mosquitoes are vectors responsible for transmitting infectious diseases such as dengue, chikungunya, and zika. For this reason, the control of these vectors through the use of larvicidal agents is vital for managing these diseases [113].

De Andrade-Porto et al. reported that the ethanolic extract of the *T. ulmifolia* leaf has an LC_50_ of 0.242 mg/mL against *Aedes aegypti* larvae in the third instar [114].

Rios et al. evaluated the larvicidal potential of EOs from different plants, either individually or in a mixture. The mixture of *T. diffusa* and *Swinglea glutinosa* EOs presented an LC_50_ against *A. aegypti* larvae between the third and fourth instars of 63 mg/L at 24 h and 34 mg/L at 48 h [115].

Duque et al. obtained similar results. They evaluated the insecticidal activity of 20 EOs against *A. aegypti* larvae between the third and fourth instars. Among the EOs evaluated, that of *T. diffusa* was among the seven oils with the highest activity, presenting a 100% mortality rate at a concentration of 100 ppm. At the same time, the LC_50_ and LC_90_ were calculated at 70 ppm and 97 ppm at 24 h, while at 48 h, they were 58 ppm and 82 ppm, respectively. Additionally, the authors determined that the EO of *T. diffusa* at a concentration equal to the calculated LC_50_ can significantly reduce the activity of enzyme targets important for insecticidal activity, such as acetylcholinesterase (47.2%), NADH oxidase (35.9%), and succinate oxidase (12.7%) [116]. It seems necessary to analyze the chromatographic profile of the essential oils to correlate the larvicidal activity with some of its components.

### 3.29. Cytotoxicity

Determining the in vitro cytotoxicity of natural products and herbal extracts is an essential step in evaluating their biological activity. These assays allow us to determine whether there are any harmful effects on healthy cells or, conversely, to identify potential therapeutic agents in cancer treatment [117]. Several authors [25,36,50,62,87] report the concentrations at which *Turnera* spp. extracts do not exert a cytotoxic effect (CC_50_ > 500 µg/mL), and therefore are safe for use. Since the last review on the bioactivity of *Turnera* spp. [4], no relevant data supporting this plant genus as a source of compounds or extracts with cytotoxic potential have been published. Even so, new publications have emerged highlighting the cytotoxic potential of secondary metabolites present in *Turnera* spp. Table 8 shows the cytotoxicity data reported in the literature for *Turnera* spp.

Kalimuthu et al. evaluated the anticancer potential of ethanolic and methanolic extract of *T. ulmifolia* callus on breast cancer cell line MCF7. The extracts exhibited a cytotoxic concentration 50 (CC_50_) > 300 µg/mL [14].

Avelino et al. determined the cytotoxicity of the methanolic extract of *T. diffusa* stem and leaf in different cancer cell lines, finding IC_50_ values in the range of 30–60 µg/mL [119]; these values coincide with those reported by Delgado et al., who obtained an IC_50_ = 50 µg/mL with the Vero cell line [22]. In the work of Avelino et al., the MDA-MB-231 cell line presented the lowest IC_50_, at 30.67 µg/mL. They also demonstrated that cell death is induced through apoptosis and necrosis. Through biodirected isolation, an active fraction was found with an IC_50_ = 32.4 µg/mL from which apigenin and arbutin **(18)** (Figure 1) were isolated; these compounds contribute significantly to the cytotoxic activity of the extract.

Gomes-Bezerra et al. evaluated the cytotoxic effect of a hydroalcoholic extract from the aerial parts of *T. diffusa* on astrocytes. The extract caused a significant increase in the percentage of cell death only at a dose of 1000 µg/mL after 6 and 24 h of exposure, while at lower doses (10 and 100 µg/mL), it had an effect similar to that of the group without treatment [121].

Velandia et al. indicated that the EO of *T. diffusa* presents a tendency towards high toxicity, with CC_50_ values of 83–199 µg/mL for different cell lines [118].

Willer et al. evaluated the cytotoxic activity of extracts and fractions of the aerial parts of *T. diffusa* in different multiple myeloma cell lines. From the fraction with the highest activity, two compounds were isolated. Naringenin **(19)** (Figure 1) (25 µM) decreased cell viability by 25% and 79% for NCI-H929 and U266 cell lines, respectively, whereas apigenin 7-O-(4″-O-p-E-coumaroyl)-glucoside **(20)** (Figure 1) (50 µM) did so by 66% and 84%, respectively. Furthermore, the isolated compounds also had a negative effect on the cell viability of peripheral blood mononuclear cells from healthy subjects [122].

Sri et al. reported that the ethanolic extract of *T. subulata* flower generates a cell death of 11.52% at 48 h in the HepG2 cell line, although the used concentration is not defined in terms of weight/volume [99].

The United States’ National Cancer Institute’s drug discovery program suggests an initial testing of extracts at a single high concentration of 100 µg/mL [123], while for isolated compounds a CC_50_ of 1–50 µM is considered active [124]. In that sense, some extracts and metabolites isolated from *T. diffusa* appear to have anticancer potential. However, this activity seems to be non-selective, since most of the extracts and compounds that presented anticarcinogenic potential also proved to be toxic in non-cancer cells. Interestingly, extracts obtained from *T. subulata* and *T. ulmifolia* showed mild cytotoxicity, with CC_50_ values above 100 µg/mL.

### 3.30. Toxicological Information

There are multiple reports in the literature regarding the toxicity of different extracts obtained from *Turnera* spp. Table 9 summarizes the toxicological information reported for different species of this genus.

Antonio et al. reported that the oral administration of 5000 mg/kg of a hydroalcoholic extract of *T. ulmifolia* in Wistar rats for 14 days did not generate signs of toxicity or an increase in body or organ weight [62].

Similarly, Gracioso et al. reported that the oral administration of up to 5000 mg/kg of an aqueous extract of *T. ulmifolia* to mice did not generate signs of acute toxicity, nor cause the death of any animal during the 14-day observation period [126].

Lopes da Costa et al. reported that the aqueous extract of *T. ulmifolia* administered orally in doses of up to 3 g/kg/day during gestation in Wistar rats did not cause histological, hormonal, or reproductive alterations. Moreover, exposure to the extract during pregnancy did not cause alterations in the development of the fetus or the embryo [129].

Bezerra et al. evaluated the acute toxicity of a hydroalcoholic extract of *T. diffusa* administered intraperitoneally (2 g/kg) and orally (5 g/kg) for 14 days. They observed that by day 3, the intraperitoneal administration of the extract caused the death of six of the ten animals. On the other hand, the oral administration group did not show signs of toxicity during the period evaluated [125].

Coe et al. reported that the aqueous extract of *T. ulmifolia* has an LC_50_ = 6650 µg/mL for *Artemia salina*, which is why it is considered nontoxic [128].

Duarte da Luz et al. reported that the oral administration of 2000 mg/kg of a hydroalcoholic extract and the ethyl acetate fraction of *T. subulata* leaves did not induce behavioral changes nor cause death in rats administered with the extracts. In addition, the authors did not observe significant differences in the hematological parameters or the weight of the organs with respect to the control group [87].

Dorantes et al. evaluated the subacute toxicity of an aqueous extract of the *T. diffusa* leaf in mice at doses of up to 1000 mg/kg for 28 days. During this time, they did not observe signs of toxicity nor changes in the behavior of the animals; no death of the animals was reported during the period evaluated. Furthermore, in the histopathological analysis, only the 1000 mg/kg dose caused slight vascular congestion in the liver of the animals. No significant differences were reported in biochemical parameters of renal and hepatic function with respect to the control [60].

Agostini et al. evaluated the toxicity of an aqueous extract of *T. hermannioides* leaves in nontarget organisms (the microalgae *Chaetoceros calcitrans* and the crustaceans *A. salina* and *Nitokra* spp.). At the maximum dose evaluated (1 mg/mL), the extract did not significantly inhibit the growth of *C. calcitrans*, and nor did it induce changes in the swimming pattern of *A. salina*. Neither did it decrease the percentage of survival of the evaluated crustaceans, so the extract was considered nontoxic [112].

Bathula et al. evaluated the acute and subacute oral toxicity of an ethanolic extract of *T. aphrodisiaca* in rats. The administration of a dose of 5000 mg/kg did not cause hematological or biochemical alterations, nor were signs of toxicity observed up to 2 weeks after the administration of the extract. The authors report an LD_50_ > 5000 mg/kg. The daily administration of doses of up to 1000 mg/kg for 4 weeks did not cause changes in the biochemical and hematological parameters evaluated, and nor did it generate signs of toxicity [127].

Rebouças et al. tested the acute toxicity of a hydroalcoholic extract of *T. subulata* flower on *A. salina* and *D. rerio*. The LD_50_ was >1000 µg/mL for *A. salina* nauplii larvae and >40 mg/kg for *D. rerio*, suggesting that the extract is nontoxic since, during the period evaluated, there was no death of the animals [24].

Despite reports of cytotoxicity of EO and extracts of *T. diffusa*, recent reports on acute and subacute toxicity show that orally administered extracts of *T. diffusa, T. ulmifolia*, and *T. subulata* do not produce harm in the different species used. Only one report of a *T. diffusa* extract administered intraperitoneally caused 60% lethality; this finding is related to the route of administration. Combining all the information, the safe use of *Turnera* spp. as medicinal plants can be concluded.

### 3.31. Genotoxic Effect

The use of medicinal plants by a large part of the population makes it necessary to study their possible toxic or harmful effects on health. It is important to determine whether the compounds present in these products cause alterations in the DNA structure and that this in turn does not generate any predisposition towards developing diseases. In this sense, genotoxicity tests help verify the safety of this type of product [130].

Senes et al. evaluated the genotoxicity of different *T. subulata* extracts on *Drosophila melanogaster* larvae using the somatic mutation and recombination test. In this work, the authors found that aqueous (5–20 mg/mL) and hydroalcoholic extracts (0.625–2.5 mg/mL), as well as the ethyl acetate fraction (0.625–2.5 mg/mL), increase the frequency of mutant spots in *D. melanogaster* exposed to the extracts for 48 h, compared with the negative control, concluding that *T. subulata* has a genotoxic effect [131].

## 4. Conclusions

The ethnopharmacological importance of the genus *Turnera* is evidenced by many reports in the literature regarding its diverse biological actions. Although *T. diffusa* is the most representative species of the genus, in the last 10 years, there has been an increased interest in studying other species of the genus, such as *T. hermannioides*, a species for which no bioactivity had been reported in the last *Turnera* review.

The number of publications per year related to the biological activity of *Turnera* spp. is shown in Figure 2. From 1998 to 2012, the average number of publications per year was 2.6. From the graph it can be noted that in the last 10 years, this average increased to almost 7. On the other hand, Figure 3 shows the number of entries related to each biological activity reported for *Turnera* spp. As there are publications that include the evaluation of more than a single class of bioactivity, Figure 3 was constructed based on the number of reports of the evaluation of a particular activity and not on the number of published articles.

It is clear that most of the available information refers to the study of the antioxidant activity of the genus, as well as the antibacterial potential. The number of entries for these two bioactivities has remained constant over the last 10 years. On the other hand, bioactivities that saw a significant increase in the number of entries were mainly antifungal, anti-inflammatory, hepatoprotective, and neuroprotective activity. In addition, there has been a growing interest in exploring other biological activities that had not been reported, such as testicular protection, insecticidal–larvicidal and immunomodulatory activity.

Regarding gastroprotective, adaptogenic, anti-obesity, or phosphodiesterase inhibitory activities, we added two results reported before 2013, which had not been included in the previous review. However, we did not find any new information related to these activities in the last ten years.

Notably, much of the available information about *Turnera*’s antioxidant activity has served as a basis for correlating other closely related bioactivities, such as hepatoprotective and neuroprotective bioactivities, and its potential to protect against testicular damage.

Finally, the need to identify the metabolites responsible for most of the biological activities reported for plants of the genus *Turnera* is worth mentioning. Of the 92 publications in this review, only 15 identify the probable metabolites involved in the reported biological activity.

The information presented in this review shows the efficacy of the plants of the genus *Turnera* as medicinal plants with a broad biological potential; in addition, the toxicity reports show the safety of its use.

## Figures and Tables

**Figure 1 pharmaceuticals-16-01573-f001:**
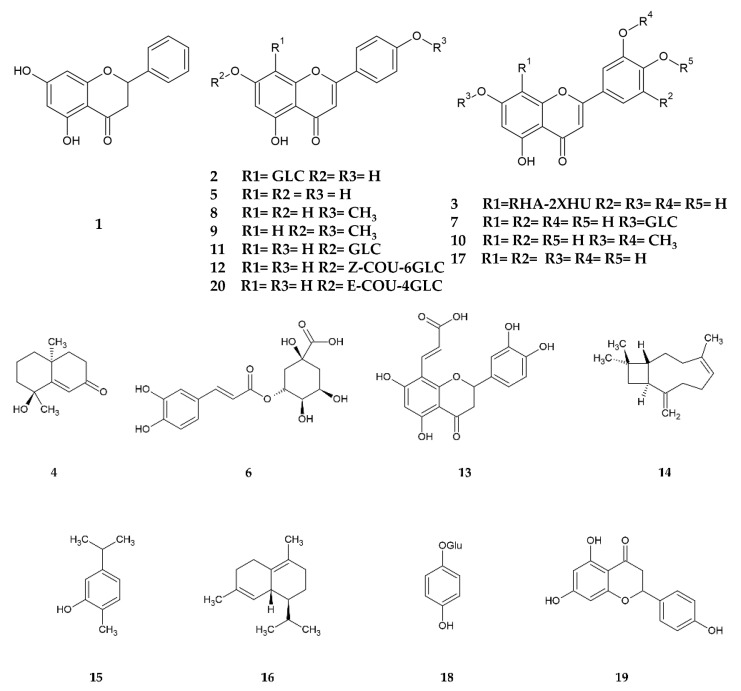
Compounds isolated from *Turnera* spp. with reported or suspected biological activity.

**Figure 2 pharmaceuticals-16-01573-f002:**
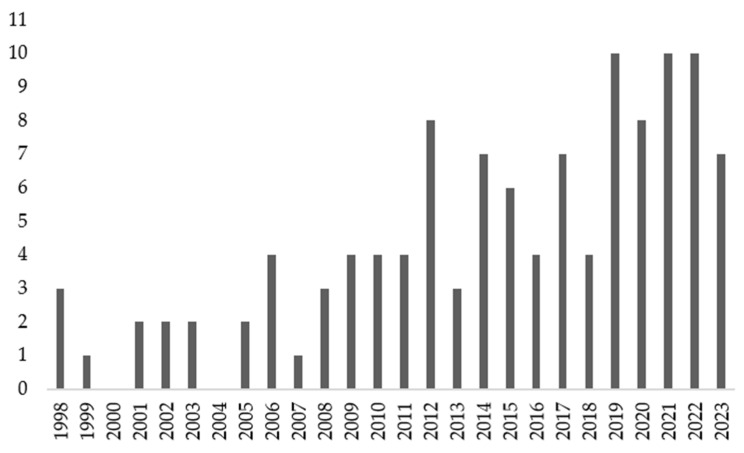
Number of publications related to *Turnera* spp. bioactivity per year (1998–2023).

**Figure 3 pharmaceuticals-16-01573-f003:**
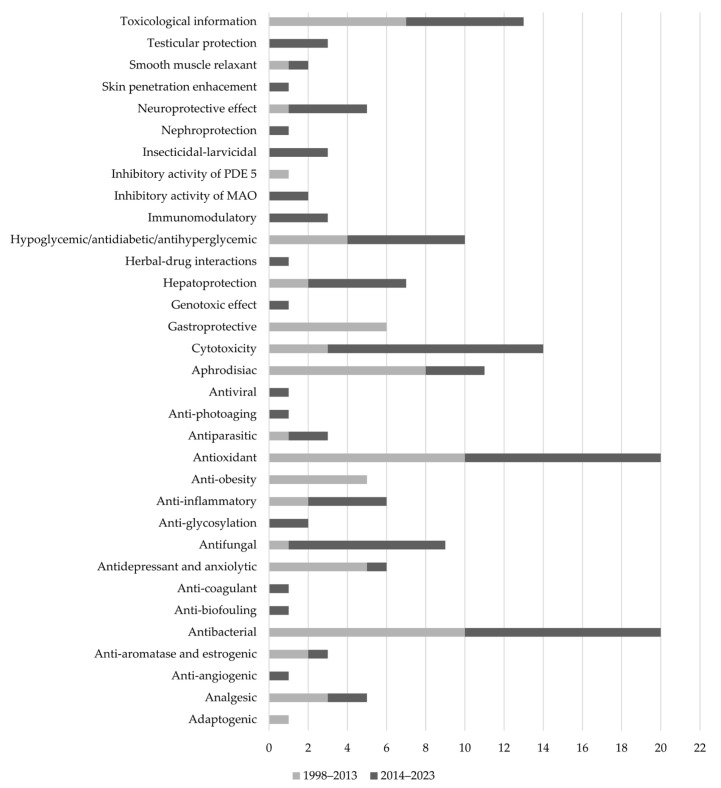
Number of entries identified for *Turnera* spp. reported bioactivities (1998–2023).

**Table 1 pharmaceuticals-16-01573-t001:** Antioxidant activity of *Turnera* spp. evaluated using free radical-based methods.

Method	Species	Part of the Plant	Extract/Compound	% Inh.	EC_50_ (µg/mL)	Ref.
ABTS	*T. diffusa*	H	HA	-	381.2	[20]
		H, T	AQ	60.8–72.3	-	[15]
		H, T	HA	30	-	[11]
	*T. subulata*	C	AE	-	250	[21]
		C	CH	-	415	[21]
		C	E	-	250	[21]
		C	HX	-	93.7	[21]
		C	M	-	500	[21]
		C	PE	-	450	[21]
DPPH	*T. diffusa*	A	Fr. AE:M	-	26.9–90.3	[22]
		A	Hepatodamianol	-	4.7	[22]
		H	E	36.2	-	[23]
		H	HA	-	305.4	[20]
		H	M	-	28.9	[19]
		H, T	A	77.9–78.1	-	[7]
		H, T	HA	78.6–77.8	-	[7]
		H, T	HA	62	-	[12]
		H, T	HA	76	-	[11]
	*T. ulmifolia*	C	AE	-	847.3	[21]
		C	CH	-	45.6	[21]
		C	E	-	10.2	[21]
		C	HX	-	991.6	[21]
		C	M	-	10.4	[21]
		C	PE	-	358	[21]
		CL	E	12.5–53.2	-	[14]
		CL	M	15.2–56.2	-	[14]
		F	E	-	25.4	[24]
		H, R, F	HA	70.6	-	[25]
H_2_O_2_	*T. subulata*	C	AE	-	120	[21]
		C	CH	-	91.2	[21]
		C	E	-	106.2	[21]
		C	HX	-	102	[21]
		C	M	-	25	[21]
		C	PE	-	425	[21]
NO	*T. subulata*	C	AE	-	31.9	[21]
		C	CH	-	24.6	[21]
		C	E	-	36.1	[21]
		C	HX	-	111	[21]
		C	M	-	25.1	[21]
		C	PE	-	338	[21]
OH•	*T. subulata*	C	AE	-	24.4	[21]
		C	CH	-	23.7	[21]
		C	E	-	21.8	[21]
		C	HX	-	30.8	[21]
		C	M	-	120	[21]
		C	PE	-	42.5	[21]
β-carotene bleaching assay	*T. diffusa*	H	M	-	233.1	[19]

Ref.: reference. A: aerial. H: leaf. C: whole plant. CL: callus. T: stem. R: root. F: flower. FR: fruit. M: methanol. HX: hexane. CH: chloroform. AE: ethyl acetate. E: ethanol. HA: hydroalcoholic. PE: petroleum ether. EO: essential oil. AQ: aqueous. % Inh: inhibition percentage. EC: effective concentration. (-): not reported.

**Table 2 pharmaceuticals-16-01573-t002:** Antioxidant activity of *Turnera* spp. reported as Antioxidant Capacity.

Method	Species	Part of the Plant	Extract/Compound	AC	Notes	Ref.
ABTS	*T. diffusa* var. *aphrodisiaca*	C	AQ	666.5	µmol ET/g	[17]
ABTS	*T. diffusa* var. *diffusa*	C	AQ	446.6	µmol ET/g	[17]
DPPH	*T. diffusa*	H	HA	145–377	mg Trolox/dry weight	[18]
NO	*T. ulmifolia*	H, R, F	HA	34.1	µM/mL nitrite formed	[25]
ORAC	*T. diffusa*	H, T	EO	813	µmol Trolox/g	[13]
FRAP	*T. diffusa*	H, T	AQ	18.5–21.3	mg GAE/g	[15]
FRAP	*T. ulmifolia*	H, R, F	HA	628.4	µM TE/g	[25]

Ref.: reference. H: leaf. C: whole plant. T: stem. R: root. F: flower. HA: hydroalcoholic. EO: essential oil. AQ: aqueous. AC: antioxidant capacity. ET: Trolox equivalents.

**Table 3 pharmaceuticals-16-01573-t003:** Antioxidant activity of *Turnera* spp. evaluated indirectly through Folin–Ciocalteu method. Results are expressed as mg gallic acid equivalents/g.

Species	Part of the Plant	Extract/Compound	Total Phenolics	Ref.
*T. diffusa*	H	AQ	9.2	[26]
		E	395.3	[18]
		HA	101.5	[20]
		M	0.4	[26]
		M	0.3	[19]
*T. diffusa*	H, T	AQ	23.9–33.8	[15]
		HA	4.7–5.4	[8]
		HA	2.5–4.7	[11]
*T. diffusa* var. *aphrodisiaca*	C	AQ	20.8	[17]
*T. diffusa* var. *diffusa*	C	AQ	8.3	[17]
*T. subulata*	C	AE	18	[21]
		CH	24.1	[21]
		E	21.2	[21]
		HX	12	[21]
		M	17.1	[21]
		PE	8	[21]
*T. subulata*	F	E	1007.4	[24]
*T. ulmifolia*	H, R, F	HA	604.8	[25]

Ref.: reference. H: leaf. C: whole plant. T: stem. R: root. F: flower. M: methanol. HX: hexane. CH: chloroform. AE: ethyl acetate. E: ethanol. HA: hydroalcoholic PE: petroleum ether AQ: aqueous.

**Table 4 pharmaceuticals-16-01573-t004:** Antioxidant activity (lipoperoxidation) of *Turnera* spp. evaluated through enzymatic methods.

Species	Part of the Plant	Extract/Compound	EC_50_ µg/mL	Inh.%	Ref.
*T. diffusa*	B	AQ	283.39	-	[27]
*T. diffusa*	H, T	HA	-	23	[12]

Ref.: reference. B: bark. H: leaf. T: stem. AQ: aqueous. HA: hydroalcoholic. EC: effective concentration. Inh.%: inhibition percentage. (-): not reported.

**Table 5 pharmaceuticals-16-01573-t005:** Antioxidant activity of *Turnera* spp. evaluated in rats.

Ref.	Species	Part of the Plant	Extract/ Compound	Dosage (mg/kg p.o.)	Agent of Damage (Dose)	Tissue Analyzed	Parameter	Levels	Notes
[28]	*T. aphrodisiaca*	H	AQ	200–400	CCl_4_ (0.7 mL/kg i.p.)	Kidney	CAT	2.7–2.9	U/mg protein
							SOD	5.9–7.1	U/mg protein
[28]	*T. aphrodisiaca*	H	E	200–400	CCl_4_ (0.7 mL/kg i.p.)	Kidney	CAT	2.4–2.5	U/mg protein
							SOD	6.9–7–6	U/mg protein
[16]	*T. diffusa*	A	HA	150	STZ (65 mg/kg i.p.)	Liver mitochondria	MDA	3.3	nmol/mg protein
							NO	0.4	µmol/mg protein
[29]	*T. diffusa*	H	-	50	Fenitrothion, K_2_Cr_2_O_7_ (68 mg/kg, 2 mg/kg)	Testicle	CAT	6	U/mg protein
							H_2_O_2_	78.7	µmol/g tissue
							GPx	1.7	U/mg
							GR	15.9	U/mg
							GSH	1.6	mmol/mg protein
							GST	0.4	µmol/h/mg protein
							SOD	51.3	U/mg protein
							TBARS	24	nmol/g tissue
[30]	*T. diffusa*	H	AQ	80	Amitriptyline (70 mg/kg i.p.)	Testicle	GPx	6.8	U/mg
							GR	16	U/mg
							GSH	2.6	mmol
							GST	0.5	µmol/h/mg protein
							SOD	67.9	U/mg protein
							TBARS	22.5	nmol/g tissue

Ref.: reference. H: leaf. A: aerial. AQ: aqueous. HA: hydroalcoholic. E: ethanolic. STZ: streptozotocin.

**Table 6 pharmaceuticals-16-01573-t006:** Antifungal activity of *Turnera* spp. evaluated by the microdilution method.

Ref.	Fungal Species	Plant Species	Part of the Plant	Extract/ Compound	MIC (µg/mL)	MFC (µg/mL)
[94]	*C. krusei* ATCC 6258	*T. diffusa*	A, R	EO	90	-
[94]	*Aspergillus fumigatus* ATCC 204305	*T. diffusa*	A, R	EO	500	-
[95]	*C. albicans*	*T. diffusa* var. *aphrodisiaca*	H	HX	50	50
				M	50	50
[95]	*C. albicans*	*T. diffusa* var. *diffusa*	H	HX	50	50
				M	50	50
[96]	*C. albicans* ATCC 90028	*T. subulata*	A	AE	310	2500
				AQ	1250	NA
				CH	630	NA
				E	1250	NA
				HX	310	2500
				M	1250	NA
[96]	*C. krusei* ATCC 6258	*T. subulata*	A	AE	160	310
				AQ	630	2500
				CH	310	630
				E	630	2500
				HX	40	160
				M	630	2500
[96]	*C. parapsilosis* ATCC 22019	*T. subulata*	A	AE	310	1250
				AQ	2500	NA
				CH	630	630
				E	1250	NA
				HX	160	630
				M	1250	NA
[96]	*C. neoformans* ATCC 90112	*T. subulata*	A	AE	80	310
				AQ	160	NA
				CH	40	160
				E	160	NA
				HX	20	160
				M	160	NA
[96]	*Trichophyton interdigitale* ATCC 9533	*T. subulata*	A	CH	630	NA
				HX	310	NA

Ref.: reference. A: aerial. H: leaf. R: root. AQ. aqueous. E: ethanolic. AE: ethyl acetate. EO: essential oil. HX: hexane. CH: chloroform. M: methanolic. MIC: minimum inhibitory concentration. MFC: minimum fungicidal concentration. NA: not active. (-): not reported.

**Table 7 pharmaceuticals-16-01573-t007:** Antibacterial activity reported for *Turnera* spp.

Ref.	Bacteria Species	Plant Species	Part of the Plant	Extract/ Compound	MIC (µg/mL)	MBC (µg/mL)
[96]	*Acinetobacter baumannii* ATCC *19606*	*T. subulata*	A	E	1250	2500
				M	1250	NA
				AE	2500	NA
[96]	*Bacillus cereus* ATCC *11778*	*T. subulata*	A	M	2500	NA
				HX	630	NA
				AE	630	NA
				CH	630	NA
[95]	*E. coli* ATCC *25922*	*T. diffusa* var. *aphrodisiaca*	H	M	50	100
				HX	50	100
		*T. diffusa* var. *diffusa*	H	M	100	100
				HX	50	100
[104]	*E. coli* MR 27	*T. subulata*	H	M	512	-
[104]	*E. coli* ATCC 25922	*T. subulata*	H	M	406	-
[96]	*E. coli* ATCC 35218	*T. subulata*	A	M	2500	2500
				HX	310	310
				AE	630	630
				CH	630	630
				E	2500	2500
[95]	*E. faecalis* ATCC 29212	*T. diffusa* var. *aphrodisiaca*	H	HX	1500	NA
				M	3000	3000
		*T. diffusa* var. *diffusa*	H	HX	1500	NA
				M	3000	3000
[95]	*K. pneumoniae* ATCC 13883	*T. diffusa* var. *aphrodisiaca*	H	HX	50	50
				M	50	100
		*T. diffusa* var. *diffusa*	H	HX	50	50
				M	50	100
[96]	*K. pneumoniae* ATCC 13883	*T. subulata*	A	HX	630	630
				CH	630	630
				AE	630	630
				E	2500	2500
				M	2500	2500
[96]	*P. aeruginosa* ATCC 27853	*T. subulata*	A	CH	630	NA
				AE	630	NA
				E	630	NA
				M	630	NA
				HX	1250	NA
				AQ	1250	NA
[104]	*P. aeruginosa* ATCC 27853	*T. subulata*	H	M	512	-
[104]	*P. aeruginosa* MR 31	*T. subulata*	H	M	512	-
[105]	*S. aureus* ATCC 11632	*T. diffusa*	H	E	300	-
[106]	*S. aureus* ATCC 25923	*T. diffusa*	-	AE	780–3120	-
[106]	*S. aureus* ATCC 33592	*T. diffusa*	-	AE	780–3120	-
[95]	*S. aureus* ATCC 25923	*T. diffusa* var. *aphrodisiaca*	H	HX	500	3000
				M	2000	2000
		*T. diffusa* var. *diffusa*	H	HX	500	3000
				M	3000	3000
[104]	*S. aureus* MR 35	*T. subulata*	H	M	512	-
[104]	*S. aureus* ATCC 25923	*T. subulata*	H	M	512	-
[96]	*S. aureus* ATCC 6538	*T. subulata*	A	HX	630	630
				CH	1250	1250
				AE	1250	2500
[107]	*S. aureus* ATCC 25923	*T. subulata*	A	EO	200	-
[107]	*S. aureus* MRSA	*T. subulata*	A	EO	25–1600	-
[107]	*S. aureus* effluxing strain	*T. subulata*	A	EO	800–1600	-
[108]	*M. fortuitum* ATCC 6841	*T. ulmifolia*	F	CH	125	-
			H	DM	125	-

Ref.: reference. A: aerial. H: leaf. F: flower. M: methanol. HX: hexane. CH: chloroform. AE: ethyl acetate. E: ethanol. EO: essential oil. AQ: aqueous. MIC: minimum inhibitory concentration. MBC: minimum bactericidal concentration. MR: multi-resistant. NA: not active. (-): not reported.

**Table 8 pharmaceuticals-16-01573-t008:** Cytotoxicity data reported in the literature for *Turnera* spp.

Method	Ref.	Species	Part of the Plant	Extract/Compound	Cell Line	CC_50_ (µg/mL)
MTT	[118]	*T. diffusa*	-	EO	B16F10	83.9 *
					B16F10	161.3 **
					HEK293	143.2 **
					HEK293	152.5 *
					HELA	109.5 **
					HepG2	186.5 *
					HepG2	199.3 **
					MCF7	139.9 **
					Vero	90.5 *
					Vero	135 **
	[94]	*T. diffusa*	A, R	EO	Vero	52.2
	[50]	*T. diffusa*	H	AQ	SH-SY5Y	>500
	[119]	*T. diffusa*	H, T	M	C-33 A	45.1
					Fibroblasts	63.24
					HepG2	43.87
					MDA-MB-231	30.67
					SiHa	50.14
					T-47D	54.02
	[120]	*T. diffusa*	H, T	EO	THP-1	>100
					Vero	88.1
	[14]	*T. subulata*	CL	E	MCF 7	>300
				M	MCF 7	>300
	[87]	*T. subulata*	H	HA	3T3	>1000
					HEK293	>1000
				Fr. AE	3T3	>1000
					HEK293	>1000
	[20]	*T. subulata*	H	HA	HaCat	100
					HDF	100
	[25]	*T. ulmifolia*	T, H, F	HA	HFF1	>1000
MTT and Alamar blue	[36]	*T. subulata*	F	AQ	RAW 264.7	>500
				HA	RAW 264.7	>500
			H	AQ	RAW 264.7	>500
				HA	RAW 264.7	>500
Neutral red uptake	[96]	*T. subulata*	A	AE	Vero	245.4
				AQ	Vero	nd
				CH	Vero	281.4
				E	Vero	nd
				HX	Vero	471.3
				M	Vero	nd
Total macromolecular content	[62]	*T. ulmifolia*	-	Fr. E	V79	465,000

Ref.: reference. CL: callus. H: leaf. T: stem. F: flower. A: aerial. R: root. AQ: aqueous. HA: hydroalcoholic. AE: ethyl acetate. E: ethanol. HX: hexane. CH: chloroform. M: methanol. EO: essential oil. nd: no significant toxicity above 640 µg/mL. CC: cytotoxic concentration. (-): not reported. *: treated after cell proliferation. **: treated prior to cell proliferation.

**Table 9 pharmaceuticals-16-01573-t009:** Toxicological information reported for *Turnera* spp.

Assays	Experimental Design	Species	Part of the Plant	Extract/ Compound	Dosage	LD_50_	Notes	Ref.
Acute Toxicity	Mice	*T. diffusa*	A	HA	2 g/kg i.p	-	Death at day 3	[125]
			A	HA	5 g/kg p.o.	-	No signs of toxicity over 14 days	[125]
			H	AQ	10–1000 mg/kg	-	No signs of toxicity or animal death, no S.D. was observed in renal and liver function parameters vs. control.	[60]
		*T. ulmifolia*	A	AQ	500–5000 mg/kg p.o.	-	There were no signs of toxicity or death. No S:D. in organ weight, feeding or feces vs. control.	[126]
			C	HA	5000 mg/kg p.o.	7.82 g/kg	-	[62]
Acute Toxicity	Rats	*T. subulata*	H	HA	500 and 2000 mg/kg p.o.	-	No changes in behavior or death of animals were observed. No S.D. was observed in hematological parameters or weight vs. control.	[87]
				Fr. EA	500 and 2000 mg/kg p.o.	-	No changes in behavior or death of animals were observed. No S.D. was observed in hematological parameters or weight vs. control.	[87]
		*T. aphrodisiaca*	H	E	5000 mg/kg p.o.	>5000 mg/kg	No signs of toxicity or changes in biochemical and hematological parameters were observed.	[127]
Subacute Toxicity	*Rattus* *norvegicus*	*T. aphrodisiaca*	H	E	250–1000 mg/kg p.o	>5000 mg/kg	No signs of toxicity or changes in biochemical and hematological parameters were observed.	[127]
Lethality assay	*Artemia* *salina*	*T. ulmifolia*	H	AQ	500–10,000 µg/mL	6650 µg/mL	-	[128]
		*T. hermannioides*	H	AQ	1 mg/mL	-	No changes in swimming pattern.	[112]
		*T. subulata*	F	E	1000 µg/mL	>1000 µg/mL	-	[24]
	*Chaetoceros calcintrans*	*T. hermannioides*	H	AQ	1 mg/mL	-	No significant growth inhibition.	[112]
	*Danio rerio*	*T. subulata*	F	E	40 mg/kg p.o.	>40 mg/mL	-	[24]
	*Nitokra sp.*	*T. hermannioides*	H	AQ	1 mg/mL	-	No decrease in survival rate.	[112]

Ref.: reference. A: aerial. C: whole plant. F: flower. H: leaf. AQ: aqueous. E: ethanolic. EA: ethyl acetate. HA: hydroalcoholic. LD: lethal dose. S.D.: significant difference.

## Data Availability

Data sharing is not applicable.

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
