# Peer review of "Bioactivity of the Genus Turnera: A Review of the Last 10 Years"

_pharmaceuticals, 2023, doi:10.3390/ph16111573_

Round 1
Reviewer 1 Report
Comments and Suggestions for Authors
Manuscript Number: pharmaceuticals-2637317
Comments to Authors
The review article “Bioactivity of the genus Turnera: a review of the last 10 years”, represents the thorough overview of genus Turnera ethnopharmacological relevance.
- Abstract: the sentence should be rewritten: “The importance of this genus, and in particular of Turnera diffusa, as a source of treatment alternatives for various conditions is evidenced by the large number of new studies that have evaluated its biological activity”.
- When the plant is first mentioned, the full Latin name, including the author name and family name, in the text should be written (page 1, line 20, and afterwards), please check throughout the text, as this refers to all plant species mentioned in the text. Regarding the bacterial strains, the first mentioning should be with full name. In addition, when listing different species of the same genus, T. should not be omitted and the short form of the plant name, with varieties should be employed (for example, page 2, line 92; page 15, line 569), please, check throughout the text and made appropriate changes…
- Page 2, line 46, 47, please rewrite, avoiding repeating the word “occurs”
- All tables should be systematically written – authors should give the date regarding the plants investigated, not the literature (example: Table 1, page 3, T diffusa , literature 9, and 78, the same part of the plant, the same type of the extracts and the same essays. The presented results are confusing and the tables are not well organised – Table 1, the used concentration of the investigated extracts are not necessary to be given in the table – if some extraordinary results appeared, that should be stressed in the text. In addition, the data represented in the tables them were not systematically organized and summarized. For example the results marked as AC, Table 1, with the meaning “antioxidant capacity” and connected to ORAC and NO tests, were not mentioned in the text, and were connected to the literature 118, and 68, that has not been mentioned in the 3.1 Section. Reference 118 was mentioned further in the text, page 27, line 907, and it did not concern the T. subulata as mentioned. Please, check all references to avoid the incorrectness when citing them.
- The antioxidant potential might be positively correlated to TP content, but FC could not be considered as method to determine the antioxidant potential, Table 2
Overall, the manuscript lacks in author’s comments on the performed up to now investigations. The impression that the manuscript represents just repeat of until now performed investigations, the narrative nature of the paper, without systematic organisation of the collected facts from numerous literature data, might decrease the scientific soundness of the review – and to avoid that the suggestion to author is to make appropriate critical comments after each mentioned biological activity, making the point if some compounds were singled as possible carriers of the discussed activity. The tables should be systematically reorganized, and references sorted.
Author Response
I am adding a document in word with the answer.

Reviewer 2 Report
Comments and Suggestions for Authors
Dear Authors,
Thank you for your work. However, I do not find your manuscript ready to be published yet. A good literature review is not easy to be prepared.
ABSTRACT
What about the most important findings? Based on the findings, can you highlight the most promising directions for the next ten years? Or - have you found some scientific boredom and poorly conducted studies? Make the Abstract eye-catching. Readers need a justified critical opinion, not only a summary of ~90 papers.
l.14
You leave some empty keyword spaces. Do you want your future paper to be readily available via search engines or not? Add at least the botanical family and most outstanding activities/types of bioassay.
INTRO
l.25
"was considered to be one of the most important therapeutic agents in ancient Mayan civilization." - Provide the reference.
SEARCH STRATEGY
Too briefly. This insufficient part projects onto the article as a whole.
1
Explain the systematic of species found in the text. Present the accepted botanical names and synonyms.
2
Add the strategy of judging what papers were useful and what were not.
3/l.38
The damianas are mainly of Latin American origin. Were any significant percent of papers in Spanish or Portuguese found in the mentioned databases? Are there any notable journals in Spanish or Portuguese that are still not indexed by the mentioned databases but you know are worthy and how to find them? Some good scientists still do not try or do not have the opportunity to publish in English.
4
What about patent activity in the field of Turnera applications? Why don't you report it? I found your patent in the References - how did you find it if only the articles and conference communications were filtered? :)
5
What about Cochrane Reviews? What about trials? The title "Pharmaceuticals" obliges.
If you do not find anything in the proposed places, just write something like "I tried here and there but found nothing special." But check and announce.
RESULTS
In my opinion, the role of authors preparing a literature review is not only to collect but also to refer to all reported papers critically.
Thus:
- Was each experiment/assay carried out according to good practices?
- Were the results supported with controls (positive and negative)?
- After reading, do you have the same opinions as the Authors of each cited paper?
- Was the plant material collected, botanically validated, and sampled correctly? What was the growth stage? How were the plants dried? ... and so on => Can we juxtapose and compare the results of similar assays?
- Were any studies connected with phytochemical studies or unreflectively evaluated the extracts' activity? Maybe add a separate table or paragraph: assays with phytochemical deep evaluation (e.g., LCMS in at least one mode).
- How do the cited authors ensure the lack/acceptable levels of organic extracting solvents? Or maybe they even didn't pay attention to this problem. Are their results still worthy to refer to?
- Were the highly lipophilic extracts (e.g., Hx) well-soluble and stable in the assay medium? Are their results still worthy to refer to?
- Are the results case reports or statistically valid studies? Add classification.
- In animal assays, put the daily dose, period of experimental therapy, and group abundances together; avoid quoting numerical data without context.
+ Subchapters often end rapidly without any summary.
=> It may be a separate table entitled "pros and cons" for collected papers. Or short conclusions/summaries below each activity-related subchapter. You can summarize your comments differently - Select one way and do it legibly and convincingly. Without this input, your manuscript is still needed and partly helpful but much less interesting.
ch.3.1
l.60
Save deleted hyphens in DPPH's full name.
l.64
"found no significant differences [...]" … under his conditions. It is a general remark for quotations.
l.87
I suggest not to mix in vitro and in vivo assays in the text.
Table 1
For a reader not involved in radical assays, explain what the results mean. Add a simple, explaining scale (e.g., from EC0-most active to EC1000-least active). Use this way of presentation not only in Tab.1 but respectively also in other subchapters.
ch.3.3
"pinocembrin, a flavonoid present in T. diffusa" - What about the concentration of this flavonoid in Turnera? Is it convincingly significant?
[31], 2016 [32], 2022
"Hepatodamianol" is not a bad name for a new drug candidate. Based on the provided structure, you are not the only one who found this molecule in nature, but it seems that your team announced this compound first (2014 vs. 2016). However, there are stereo-inconsistencies with your own patent and papers presentations of "hepatodamianol" (see attached pdf file). Thus, I expect you first to clear the problem with the structure presentation. Next, when you decide what "hepatodamianol" is, I wish you to at least report the other sources of this cpd and closely related C-flavones and explain the doubled common name problem. See "isocassiaoccidentalin B," CAS No 2019182-05-7, DOI: 10.1002/hlca.201600131. The yield of "hepatodamianol"/"isocassiaoccidentalin B" from Turnera and Cassia is a separate issue to be discussed if you want to name it a genus/species marker. By the way, "hepatodamianol" is still not recorded by CAS.
In my opinion, the lack of a general phytochemical part in the Introductory part of this review makes it less informative.
In the following activity-related subchapters, follow general Results suggestions.
An example of an unacceptable referring style:
l.403
"Moreover, Dorantes et al. found that daily administration for 28 days of 100 and 1000 mg/kg of an aqueous extract of T. diffusa leaves to mice significantly decreased body weight [54]." - Who was treated with the therapy? What was the population? What was the concentration of "plant in extract" (defined by, e.g., DER/DSR)? Why to present doses if the above are hidden?
l.901
"guarantee" - I cannot agree; maybe "make us more convinced"; avoid emotionally charged opinions / large quantifiers in scientific language.
CONCLUSIONS
This chapter is dramatically laconic.
l.927
Avoid the use of any abbreviations that were not explained before. Correct it.

English is fine. Thank you.
Author Response
I am loading a document in word with the answer

Reviewer 3 Report
Comments and Suggestions for Authors
The study of Aída Parra-Naranjo et al. aimed to review the updated biological effects of species in the genus Turnera in the last 10 years.
The manuscript is informative to readers, and in accordance with the aim and scope of Pharmaceuticals.
However, there is some information that needs to improve to make the manuscript more informative, including:
1. Lines 229, 348, 428 …: Many bioactive compounds from the genus Turnera were mentioned; however, the structures of these compounds were lacking in the manuscript. Thus, I highly recommend the addition of the structures of these compounds for clarity for the reader.
2. Conclusion section: Data Analysis. The authors list many pharmacological properties of species in the genus Turnera but fail to assemble the data and draw meaningful (data driven) conclusions. For example, “a column graph indicating publications per year over the last 10 years” and “a circle graph indicating the % of each bioactivity in the last 10 years” would have been very helpful.
Author Response
I am uploading a document i word with the answers

Round 2
Reviewer 1 Report
Comments and Suggestions for Authors
The authors improved their manuscript.
Reviewer 2 Report
Comments and Suggestions for Authors
Dear Authors,
Thank you for providing the improved version.
Concerning a review as a type of manuscript, I understand your position and I know that the reviews of a similar construction to your one are published. This is the role of the Editors to decide what they accept or not, not mine. Thus, today I vote to proceed the manuscript At the same time, I encourage you to delve deeper into the topic. Tomorrow, database engines will do such juxtapositions automatically. Concerning the auto-reviews in the future, the robots will only reproduce the quality they have been taught by the real reviewers. Thus, it is your role to give the quality, today.
Concerning the patent doubts, I do not see anything against referring to patent, even Authors patent. I just referred to the Strategy part in which the patent search was absent (today l.60-61).
Concerning the stereochemical doubts of cpd 3, I do not have any personal objections against you. I sent you enough material to think. I agree, my message was not directly connected to Damiana papers and you may have the reason to ommit some topics. However, on the basic of accidentally found documents and similar or identical compounds, I encourage you to register the proper version of your molecule in the Chemical Abstracts.
Intro, l.50+
This is the criticism suitable for the Results and Discussion but not to the Intro.
Tables
Preparing the final version, avoid last single lines moving to the next page.